



# Exploring Sources of Ice Crystals in Cirrus Clouds: Comparative Analysis of Two Ice Nucleation Schemes in CAM6

Kai Lyu[1], Xiaohong Liu[1], Bernd Kärcher[2]

[1]Department of Atmospheric Sciences, Texas A&M University, College Station, 77840, USA
[2]DLR Institut für Physik der Atmosphäre, Oberpfaffenhofen, Wessling, 82234, Germany

*Correspondence to*: Xiaohong Liu (xiaohong.liu@tamu.edu)

**Abstract.** Ice nucleation, a critical process in cirrus clouds, remains a challenge in global climate models. To enhance the understanding, a novel ice nucleation parameterization based on the Kärcher (2022) (K22) scheme is introduced into the NCAR Community Atmosphere Model version 6 (CAM6).

To investigate ice formation in cirrus clouds, sensitivity tests are conducted to analyze three ice sources: orographic gravity wave (OGW)-induced, convection detrained and turbulence-induced. These tests employ both the K22 scheme and the default Liu and Penner (2005) (LP05) scheme. Model evaluation includes 6-year climatology and nudged simulations representing the Small Particles in Cirrus (SPARTICUS) and O2/N2 Ratio and CO2 Airborne Southern Ocean Study (ORCAS) campaigns.

The climatology simulations reveal that both schemes concentrate detrained and turbulence-induced ice crystals in low to mid-latitudes, whereas OGW-induced ice crystals are concentrated in mid- to high latitudes. Compared to the LP05 scheme, the K22 scheme generates a higher number of ice crystals. The simulated cloud microphysical properties using the K22 scheme align well with observations for orographic cirrus during the SPARTICUS campaign.

In orographic cirrus, both schemes identify OGW-induced ice crystals as the dominant ice source. However, due to
distinct competition parameterizations, the K22 scheme exhibits less competition from minor ice sources (convection detrained and turbulence-induced). This underscores the significance of competition mechanisms within nucleation schemes for accurate cirrus clouds simulation. The application of two distinct nucleation schemes provides valuable insights into the dominant ice sources in cirrus clouds.





## 1. Introduction

Cirrus clouds play an important role in the Earth's radiation budget, thereby affecting the climate (Liou, 1986). These ice clouds can reflect solar radiation back to space, cooling the planet. They can also absorb terrestrial longwave radiation and thus warm the planet. The balance between these two opposite processes is greatly influenced by the microphysical properties of ice crystals in cirrus clouds and determines the net cloud radiative forcing. The representation of cirrus clouds in global climate models (GCMs) has been recognized as a key factor in understanding the climate change (Boucher et al., 2013).

Ice crystals in cirrus clouds originate from two main processes, detrainment from convective clouds and in-situ nucleation (Krämer et al., 2016; Muhlbauer, Ackerman, et al., 2014). Cirrus clouds are formed through convective detrainment when air containing ice crystals flows out of convective clouds, such as anvils. These clouds are usually associated with high ice number concentrations ($> 100\ \mathrm{L^{-1}}$) (Heymsfield et al., 2017).

Ice crystals in in-situ cirrus clouds are primarily nucleated by aerosols. There are two nucleation mechanisms: homogeneous freezing of solution droplets and heterogeneous nucleation on ice nucleating particles (INPs). Homogeneous nucleation requires higher supersaturation ($> \sim40\text{-}60\ \%$) and lower temperatures ($< -37\ ℃$), typically resulting in high ice number concentrations ($> 100\ \mathrm{L^{-1}}$). In contrast, heterogeneous nucleation occurs at lower supersaturation and higher temperatures, involving INPs such as dust and black carbon (BC). This process generally produces low ice number concentrations ($< 10\ \mathrm{L^{-1}}$).

Substantial progress has been made in understanding homogeneous nucleation (Koop et al., 2000). Homogeneous nucleation is usually triggered by high vertical velocies ($> 0.1\ \mathrm{m\ s^{-1}}$). These dynamic factors can be induced by either turbulence in the unstable circumstances with small Richardson numbers or gravity waves in the stable atmosphere with large Richarson numbers (Heymsfield et al., 2017).

Recent studies on cirrus clouds in GCMs usually overlook the roles of ice crystal sources, especially for cirrus clouds with high ice number concentrations ($> 100\ \mathrm{L^{-1}}$). The absence or misrepresentation of a critical ice source may lead to the failure to simulate cirrus cloud properties. For example, most GCMs treat turbulence as the sole vertical velocity mechanism driving ice nucleation. However, research has shown that due to limitations in higher-order turbulence closure theory, cirrus

clouds formed by gravity waves are usually absent in GCMs (Golaz et al., 2002b; Huang et al., 2020). Notably, studies have

demonstrated that incorporating the effects of orographic gravity waves (OGWs) into ice nucleation processes enables

models to successfully simulate the observed characteristics of orographic cirrus clouds (Lyu et al., 2023). In addition, many

studies highlight that ice crystals from convective detrainment can have a significant impact on the microphysical properties

of cirrus clouds, particularly in the tropical regions (Horner & Gryspeerdt, 2023; Horner & Gryspeerdt, 2024; Nugent et al.,

2022). In this study, we focus on three ice sources: OGW-induced, turbulence-induced and convective detrained.

Significant uncertainties persist regarding heterogeneous nucleation (Hoose & Möhler, 2012; Murray et al., 2012) and

the interplay between the two nucleation mechanisms (Kärcher et al., 2022; Shi et al., 2015). There are many uncertainties in

the nucleation processes and can complicate the prediction of cirrus clouds microphysical properties (Knopf & Alpert, 2023).

Moreover, uncertainties regarding the number concentration and chemical composition of INPs in the upper troposphere

along with their nucleation mechanisms further hinder cirrus cloud simulations (DeMott et al., 2011).

Several parameterizations of nucleation mechanisms have been developed in GCMs. Liu and Penner (2005) (LP05)

developed a parameterization that includes homogeneous nucleation, heterogeneous nucleation and their interactions. The

parameterization was subsequently applied to the NCAR Community Atmospheric Model (CAM) (Liu et al., 2007) and was

further refined to include the effects of pre-existing ice (Shi et al., 2015).

A new parameterization (Kärcher, 2022), referred to as K22, that encompasses homogeneous nucleation, heterogeneous

nucleation, their interactions, and competition with preexisting cirrus ice, has been integrated into CAM6. The purpose of

this paper is to integrate the K22 nucleation parameterization into GCMs and evaluate its effects on cloud microphysical

properties and dominant sources of ice crystals in cirrus clouds. Section 2 presents a description of the model, and the

parameterization method used in the study. The observational data employed for evaluation are described in Section 3. The

model results, along with comparisons to the default LP05 parameterization, are discussed in Section 4. Finally, the

summary and conclusions are presented in Section 5.



## 2. Model and Parameterization

### 2.1 Model Description

The NCAR Community Atmosphere Model version 6 (CAM6) model is the atmosphere component of Community Earth System Model version 2 (CESM2) (Danabasoglu et al., 2020). CAM6 employs the updated Morrison-Gettelman cloud microphysics scheme (MG2) to predict the mass and number concentrations of cloud liquid, cloud ice, rain and snow (Gettelman & Morrison, 2015; Morrison & Gettelman, 2008). The deep convection processes are represented using the Zhang and McFarlane (1995) scheme. The planetary boundary layer turbulence, cloud macrophysics, and shallow convection are treated by the Cloud Layers Unified by Bi-normals (CLUBB) (Bogenschutz et al., 2013; Golaz et al., 2002a; Hinz et al., 1996). Aerosols are treated using the 4-mode version of Modal Aerosol Model (MAM4) (Liu et al., 2016). Since CLUBB effectively represents turbulence with a small Richardson number but struggles to produce perturbations caused by gravity waves (Golaz et al., 2002a, 2002b; Huang et al., 2020), subgrid-scale orographic gravity waves (OGWs) and turbulence kinetic energy (TKE) from CLUBB (CLUBB-TKE) are introduced into the ice nucleation as sources of vertical velocity (Lyu et al., 2023). Aerosols involved in ice nucleation act interactively with the MAM4. When new ice crystals form, the nucleated aerosols are transferred from the interstitial state to the cloud-borne state. Similarly, when cloud droplets form, the nucleated aerosols are transferred to the cloud-borne state and are subject to precipitation scavenging. The radiation calculations are based on the Rapid Radiative Transfer Model for General Circulation Models (RRTMG) (Iacono et al., 2008). The climatology experiments and nudged simulations related to the Small Particles in Cirrus (SPARTICUS) and O2/N2 Ratio and CO2 Airborne Southern Ocean Study (ORCAS) campaigns are designed and listed in Table 1 and 2. All simulations are conducted at a resolution of 0.9º × 1.25º with 56 vertical layers.

The LP05 ice nucleation scheme involves two mechanisms: homogeneous and heterogeneous nucleation (Liu & Penner, 2005). Developed base on numerical parcel model simulations, this scheme considers the competition between homogeneous and heterogeneous nucleation processes, as well as their interactions with pre-existing ice crystals (Shi et al., 2015). To fit to observed ice number concentrations (Gettelman et al., 2010), homogeneous nucleation utilizes sulfate aerosols in the Aitken mode with diameters greater than 0.1 μm.





In our study, the OGW experiments serve as the reference experiments. These experiments consider three primary sources of ice crystals: convective detrainment, nucleation driven by turbulence (CLUBB-TKE), and nucleation driven by OGWs. To isolate the effects of each source, we designed three sensitivity tests: no_DET, no_TKE and no_OGW, each excluding one of these specific sources. By comparing the differences in ice number concentration ($N_i$) between the

reference experiments and sensitivity experiments, we can aim to understand the contribution of each ice source in CAM6.

**Table 1 Description of 6-year Climatology Simulations**

| Model experiment | Description |
| --- | --- |
| LP05_OGW-Climo | Default CAM6 configuration with turbulence (CLUBB-TKE) and orographic gravity waves (OGWs) for ice nucleation. |
| LP05_no_OGW-Climo | Same as LP05_OGW-Climo but without OGWs for ice nucleation |
| LP05_no_DET-Climo | Same as LP05_OGW-Climo but without detrained ice. |
| LP05_no_TKE-Climo | Same as LP05_OGW-Climo but without turbulence for ice nucleation. |
| | |
| LP05_OGW-Homo-Climo | Same as LP05_OGW-Climo but only consider homogeneous ice nucleation. |
| LP05_OGW-Hete-Climo | Same as LP05_OGW-Climo but only consider heterogenous ice nucleation. |
| K22_OGW-Climo | Same as LP05_OGW-Climo but with K22 nucleation parameterization. |
| K22_no_OGW-Climo | Same as LP05_no_OGW-Climo but with K22 nucleation parameterization. |
| K22_no_DET-Climo | Same as LP05_no_DET-Climo but with K22 nucleation parameterization. |
| K22_no_TKE-Climo | Same as LP05_no_TKE-Climo but with K22 nucleation parameterization. |
| K22_OGW-Homo-Climo | Same as K22_OGW-Climo but only consider homogeneous ice nucleation. |
| K22_OGW-Hete-Climo | Same as K22_OGW-Climo but only consider heterogenous ice nucleation. |
| K22_OGW_Shan-Climo | Same as K22_OGW-Climo but with aerosol wet removal in convection (Shan et al., 2021). |




**Table 2 Description of Nudged Simulations**

| Model experiment | Description |
| --- | --- |
| **2009 October to 2010 June** | |
| LP05_OGW-SP | Default CAM6 configuration with turbulence and orographic gravity waves (OGWs) for ice nucleation. |
| LP05_no_OGW-SP | Same as LP05_OGW-SP but without OGWs for ice nucleation |
| LP05_no_DET-SP | Same as LP05_OGW-SP but without detrained ice. |
| LP05_no_TKE-SP | Same as LP05_OGW-SP but without turbulence for ice nucleation. |
| K22_OGW-SP | Same as LP05_OGW-SP but with K22 nucleation parameterization. |
| K22_no_OGW-SP | Same as LP05_no_OGW-SP but with K22 nucleation parameterization. |
| K22_no_DET-SP | Same as LP05_no_DET-SP but with K22 nucleation parameterization. |
| K22_no_TKE-SP | Same as LP05_no_TKE-SP but with K22 nucleation parameterization. |
| K22_OGW-Homo-SP | Same as K22_OGW-SP but only consider homogeneous ice nucleation. |
| K22_OGW-Hete-SP | Same as K22_OGW-SP but only consider heterogenous ice nucleation. |
| **2015 October to 2016 February** | |
| LP05_OGW-OR | Same as LP05_OGW-SP except simulation period. |
| LP05_no_OGW-OR | Same as LP05_no_OGW-SP except simulation period. |
| LP05_no_DET-OR | Same as LP05_no_DET-SP except simulation period. |
| LP05_no_TKE-OR | Same as LP05_no_TKE-SP except simulation period. |
| K22_OGW-OR | Same as K22_OGW-SP except simulation period. |
| K22_no_OGW-OR | Same as K22_no_OGW-SP except simulation period. |
| K22_no_DET-OR | Same as K22_no_DET-SP except simulation period. |
| K22_no_TKE-OR | Same as K22_no_TKE-SP except simulation period. |

## 2.2 K22 Ice Nucleation Parameterization

In the K22 parameterization, homogeneous freezing is treated as a stochastic process in which the number of solution

droplets decreases over time based on freezing rate. The freezing rate is determined using the liquid water volume of the

droplet population and a rate coefficient derived from a water activity-based formula (Koop et al., 2000). Vertical velocity



significantly influences water activity (Baumgartner et al., 2022; Kärcher et al., 2022; Liu & Penner, 2005). The scheme

assumes a monodisperse liquid solution droplet distribution at a wet radius of 0.25 μm. The formulation of the number of ice

crystals nucleated homogeneously is described by Kärcher et al. (2022).

A deterministic (time-independent) approach to predict the number ($n$) of activated INPs is employed in the K22

parameterization as follows:

$$n = n_{tot}\Phi(s), \tag{1}$$

where $n_{tot}$ is the total INP number concentration and $\Phi$ is the activated INP fraction. $\Phi$ can be represented as either a linear

ramp or a hyperbolic tangent function. Since we consider dust as the INPs, a linear ramp is applied in our study.

The function $\Phi$ can be expressed as follows:

$$\Phi = \begin{cases} 0 & : \ s < s_{min} \\ \frac{(s-s_{min})}{s_{max}-s_{min}} & : s_{min} \le s \le s_{max}, \\ 1 & : \ s > s_{max} \end{cases} \tag{2}$$

where $s_{min}$ and $s_{max}$ are two parameters that define the range of ice supersaturation where heterogeneous nucleation can

occur. In our study, they are set to 0.22 and 0.3, respectively.

The equation governing the temporal evolution of ice supersaturation, $s$, in the ice-vapor system is expressed as

$$\frac{ds}{dt} = a(s+1)w - \int_0^s \frac{4\pi}{vn_{sat}} \frac{dn}{ds'} \left( \int_{\tau(s')}^{t(s)} r^2 \frac{dr}{dt} dt \right) ds' , \tag{3}$$

where $\frac{ds}{dt}$ represents the time derivative of $s$. The first term on the right-hand side of the equation is the production term

related to adiabatic cooling. $a$ is a thermaldynamic parameter (Pruppacher et al., 1998) relating to adiabatic vertical air

motion, and $w$ is restricted to the updraft speed ($w > 0$). The second term signifies the loss term due to the removal of water

vapor. The upper integration limit is the time $t$ corresponding to ice supersaturation $s$, and the lower integration limit is a

time $\tau$ corresponds to $0 \le s' \le s$.

Within the integral, $r$ is the radius of spherical ice crystals, $\frac{dr}{dt}$ denotes the associated growth rate per ice crystal, $v$

represents the volume of one water molecule in bulk ice, and $n_{sat}$ is the water vapor number concentration in gas phase at ice

saturation. The number concentration of ice crystals formed by INPs in a range of supersaturation d$s'$ is given by $\frac{dn}{ds'}$.





The loss term in Equation (3) can be integrated numerically as described by Kärcher (2022). When $\frac{ds}{dt} = 0$, we can

estimate the total heterogeneously nucleated ice number concentrations. Quenching velocities $w_q$ are defined as:

$$w_q = \frac{Loss\ term\ in\ Equation\ (3)}{a(s+1)},\qquad(4)$$

where the loss term includes contributions from heterogeneous nucleation and pre-existing ice. This approach allows us to

determine an effective vertical updraft $w_{eff}$ which is used to describe conditions relevant to the homogeneous nucleation.

The effective vertical updraft speed $w_{eff}$ is calculated as:

$$w_{eff} = w - w_{q,het} - w_{q,pre},\qquad(5)$$

where $w$ is the updraft speed, $w_{q,\,het}$ is the quenching velocity for ice crystals due to heterogeneous nucleation, and $w_{q,pre}$ is the

quenching velocity due to pre-existing ice. If $w_{eff} \leq 0$, no homogeneous freezing occurs. When $w_{eff} > 0$, homogeneous

nucleation will take place, but homogeneously nucleating ice number concentration will be smaller than that in the absence

of INP-derived and pre-existing ice crystals (i.e. that calculated based on $w$).

There are some differences in the competition with pre-existing ice between the K22 and LP05 nucleation schemes.

Uncertainties exist regarding the relationship between the reduction of supersaturation caused by pre-existing ice crystals

and the number of homogeneously nucleated ice crystals suppressed by pre-existing ice crystals. In the K22 parameterization

quenching velocity, which is determined using a loss term associated with pre-existing ice crystals, is employed to calculate

the number of homogeneously nucleated ice crystals.

However, the LP05 parameterization estimates the homogeneously nucleated ice number from a different perspective.

The scheme first assumes the absence of pre-existing ice crystals and calculates the ice number that would be generated

homogeneously. The calculated ice number is then compared the pre-existing ice number. If the pre-existing ice number is

smaller than the calculated number, additional ice crystals are generated homogeneously, with the resulting ice crystal count

equal to the difference. Otherwise, no new ice crystals are nucleated.

Since the K22 scheme assumes spherical ice crystal shapes, it may underestimate the surface area available for vapor

deposition. This underestimation can exaggerate when pre-existing ice crystals are small and numerous. This bias may lead

to an underestimation of supersaturation depletion by these ice crystals, potentially facilitating the occurrence of

homogeneous nucleation. However, the LP05 scheme emphasizes competition between newly nucleated and pre-existing ice




crystals, influenced by the large size of model grid. If a small fraction of the grid contains high concentrations of pre-existing

ice crystals, the LP05 scheme can suppress new ice formation even when new clouds form elsewhere within the same grid.

## 3. Observational Data

### 3.1 SPARTICUS campaign

This study utilizes observational data obtained during the SPARTICUS field campaign, conducted from January to June

2010 in the Central United States. The flight tracks of the campaign are depicted in Fig. 1a, covering approximately 150

165    research flight hours targeting cirrus clouds. Temperature measurements were conducted using the Rosemount probe Model

102 probe with a precision of ±0.5 ℃. Vertical velocity was measured by the Aircraft-Integrated Meteorological

Measurement System-20 (AIMMS-20) instrument mounted on a Learjet 25 (Muhlbauer, Kalesse, et al., 2014). Ice crystals in

the size range of 10–3000 µm were measured using two-dimensional stereo-imaging probes (2D-S). The 2D-S probe

minimizes biases in the number concentration of small-sized ice crystals by addressing ice shattering effects (Lawson, 2011).

Observational data were sampled at a frequency of 1 Hz.



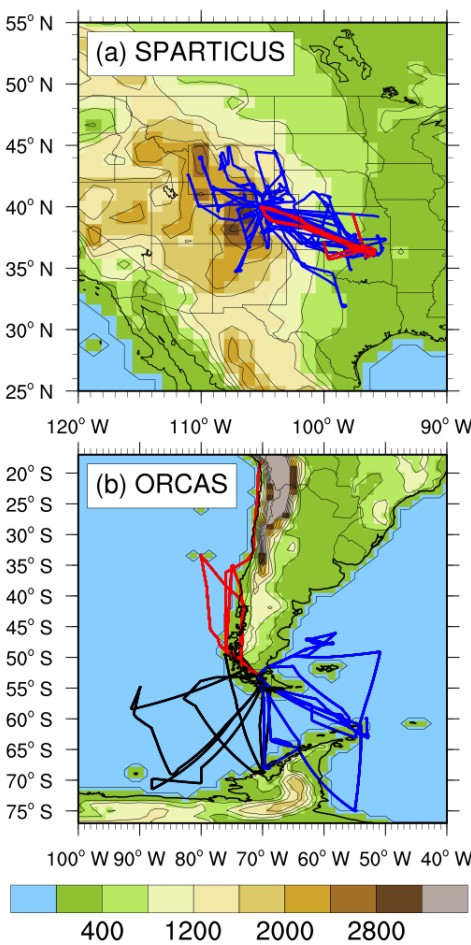

**Figure 1. The top panel (a) shows aircraft trajectories (solid blue lines) during the SPARTICUS campaign. Solid red lines indicate flight tracks on days when orographic cirrus was observed (March 19, 30, April 1, 28, and 29, 2010). The bottom panel (b) shows aircraft trajectories during the ORCAS campaign. Color shading and black line contours illustrate the surface terrain (in m). Red lines denote flight tracks in Region 1, located north of Punta Arenas, Chile (SCCI), on the following days: January 23, and 25, February 8, 10, 17, 19, 22, 23 and 29, 2016.  Blue lines denote flight tracks in Region 2, southeast of SCCI, on January 18, 25, and 30, February 12, 18, and 25, 2016. Black lines show flight tracks in Region 3, southwest of SCCI, on January 15, and 21, February 5 and 24, 2016.**

At a speed of approximately 230 m s$^{-1}$, the aircraft covers about 100 km in 430 seconds of flight time, which corresponds to the model's horizontal resolution (1 degree). To facilitate a meaningful comparison between observational data and model outputs, a running average of 430 seconds of measurement data is applied (Patnaude et al., 2021). Additionally, the microphysical properties (such as ice number $N_i$, ice water content IWC and number-weighted diameters

$D_{num}$) of ice crystals with diameters larger than 10 µm from CAM6 results are derived using the size cut method described by

Eidhammer et al. (2014), consistent with the measurements obtained by the 2D-Stereo Particle Probe (2D-S).

## 3.2 ORCAS campaign

The $O_2/N_2$ Ratio and $CO_2$ Airborne Southern Ocean Study (ORCAS) was an NSF-sponsored airborne field campaign

conducted from Chile during January and February 2016. The campaign utilized the NSF/NCAR HIAPER Gulfstream V

(GV) aircraft for 18 flights over a period of 6 weeks. The data, sampled at 1 Hz, encompasses a total of 95 flight hours

(Stephens et al., 2018). Ice cloud particles are measured by the Fast 2-Dimensional Optical Array Cloud probe (Fast-2DC),

which detects particle sizes ranging from 62.5 to 1600 µm (excluding the first two bins due to the ice shattering effects). The

primary difference in measuring ice properties between the SPARTICUS and ORCAS campaigns is the instrumentation used

to measure ice crystals. The SPARTICUS campaign employs the Fast 2D-S probe, while the ORCAS campaign utilizes the

2D-C probe. Due to the ice shattering effect, the reliability of small ice measurements is compromised with the 2D-C probe.

The subsequent paragraphs will delve into ice microphysical properties, specifically focusing on large-size ice crystals ($D_{num}$

$\geq 62.5\mu m$) observed during the ORCAS campaign.

The ORCAS flight profiles encountered a lot of samples of cold upper-tropospheric clouds. To derive the properties

(such as $N_i$, IWC and $D_{num}$) of ice crystals with diameter $\geq 62.5$ µm from CAM6 results, the size cut method described by

Eidhammer et al. (2014) is employed. This methodology ensures consistency with the measurements obtained by the 2D-C

probe (Section 3.1).

To better evaluate the model results, this study divides the ORCAS flights into three regions, as illustrated in Fig. 1b.

Flights in Region 1 primarily traverse high mountain ranges where cirrus clouds form primarily due to OGWs, together with

convection and frontal waves. Flights spanning Regions 2 and 3 predominantly cover oceanic areas, heavily influenced by

convection and frontal waves. Notably, Region 2 is located downwind of the Andes Mountains and Antarctic high plateaus,

thereby experiencing the additional influence from OGWs on observed cirrus cloud microphysical properties.

This regional division allows for a more detailed analysis of cirrus cloud processes. The observed differences in cloud

microphysical properties across these three regions highlight the distinct characteristics of cirrus clouds over land and ocean,





particularly in mid- and high latitudes. These differences can provide insights into how various ice nucleation processes and environmental factors influence cirrus clouds formation and evolution.

**4. Results**

**4.1 Climatology Experiments**

Fig.2 illustrates the grid-mean ice number concentration ($N_i$) for different types of cirrus in climatology experiments using the LP05 and K22 schemes. The results indicate that $N_i$ is generally higher in the K22_OGW-Climo experiment compared to the LP05_OGW-Climo experiment. In both schemes, ice crystals detrained from convection are primarily

concentrated in the tropical regions and mid-latitudes. Orographic cirrus due to OGWs are concentrated over mid- and high-latitudes, and in situ nucleated ice crystals induced by turbulence are prevalent at the top of tropical tropopause layers (TTL) and in mid-latitudes. Across all three ice sources, experiments based on the K22 scheme produce higher ice number concentrations than those based on the LP05 scheme, mainly from the OGW-induced cirrus.



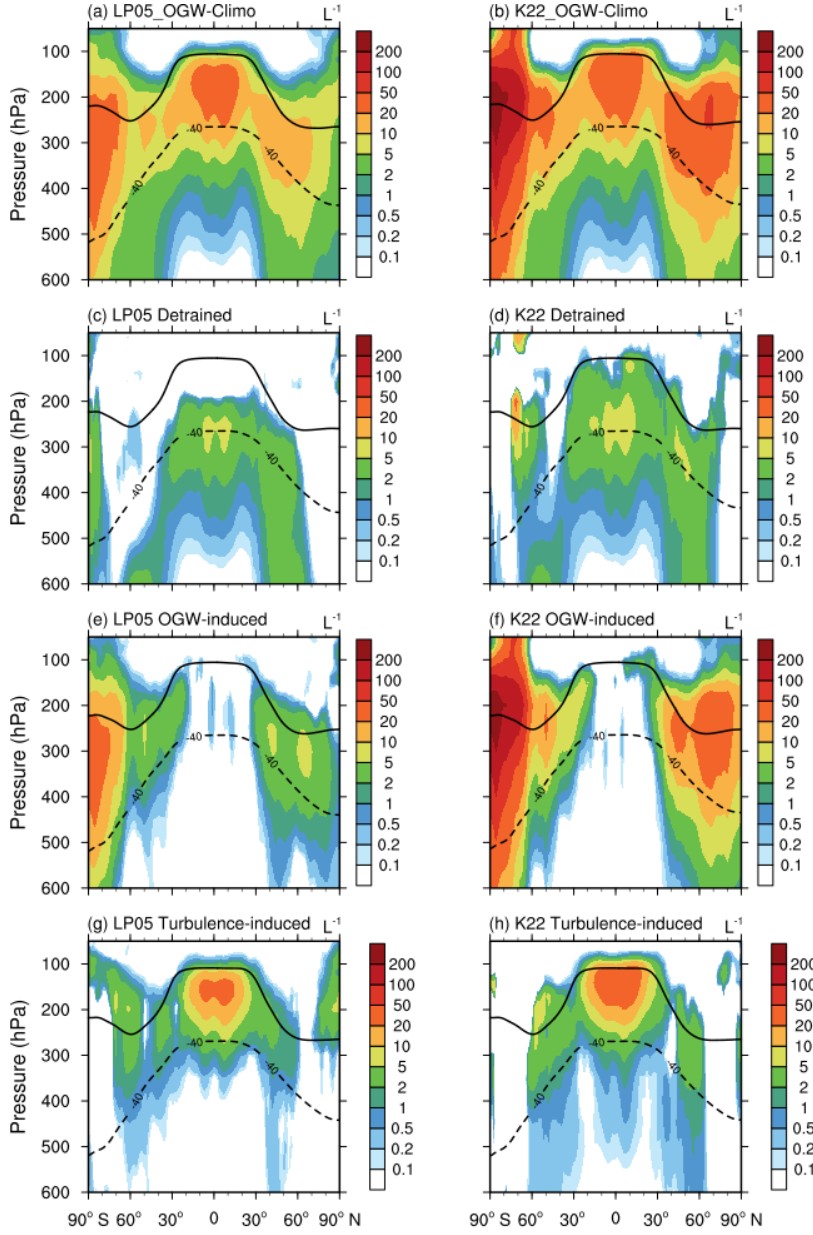

**Figure 2. Annual zonal grid-mean ice number concentration ($N_i$) from 6-year climatology simulations in the upper troposphere (above 600 hPa). The first row shows $N_i$ from the LP05_OGW-Climo and K22_OGW-Climo experiments. The second row shows the differences in $N_i$ between OGW and no_DET experiments (OGW – no_DET) for both the LP05 and K22 schemes, highlighting the contribution from cirrus clouds associated to convective detrainment. The third row presents the $N_i$ differences between OGW and no_OGW experiments (OGW – no_OGW) for both schemes, indicating the presence of orographic cirrus. The fourth row presents the $N_i$ differences between OGW and no_TKE experiments (OGW – no_TKE) for both schemes, reflecting cirrus clouds formed due to turbulence. Dashed lines represent the annual mean -40°C isothermal line, while solid lines indicate the tropopause in the corresponding simulations.**





We further analyze grid-mean $N_i$ in the sensitivity tests using homogeneous-only and heterogeneous-only experiments

(shown in Fig. 3). These experiments include OGW-induced, turbulence-induced and detrained sources of ice crystals. The

results reveal that both nucleation processes produce more ice crystals in the K22 scheme compared to the LP05 scheme. In

addition, the $N_i$ results from the OGW-Climo experiments in both the K22 and LP05 schemes closely resemble those from

their corresponding OGW-Homo-Climo experiments. This similarity indicates that homogeneous nucleation is a major

contributor to the nucleated ice number globally in both the LP05_OGW-Climo and K22_OGW-Climo experiments.


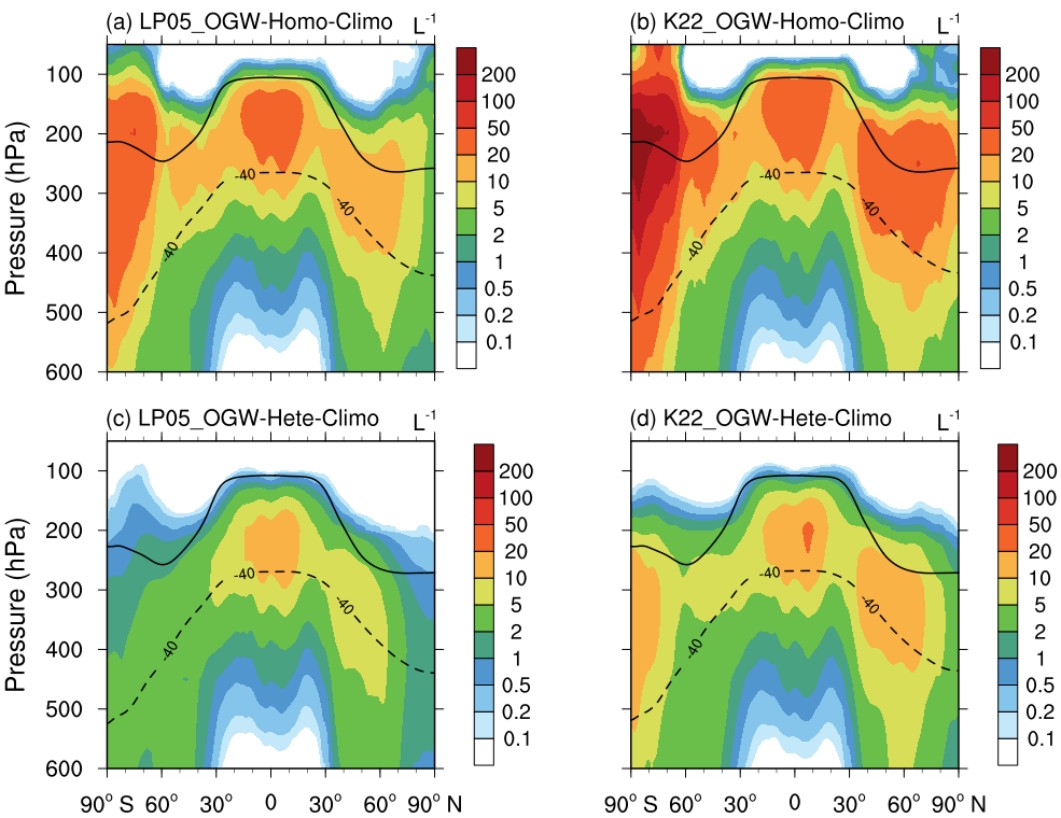

**Figure 3. Annual zonal grid-mean $N_i$ from 6-year Climatology simulations in the upper troposphere (above 600 hPa). Dashed lines indicate the annual mean -40 °C isothermal line, and solid lines represent the tropopause in the**
**corresponding simulations.**



The higher activated number concentration of aqueous aerosols for homogeneous nucleation in the K22 scheme, compared to the LP05 scheme, can be attributed to both direct and indirect influences. Although the homogeneous nucleation parameterizations in both schemes are based on Koop et al. (2000), the direct influence on homogeneously nucleated ice number arises from differences in the competition with pre-existing ice in the two schemes. As illustrated in Section 2.2, the LP05 scheme tends to encourage competitions between pre-existing ice crystals and newly formed ice crystals, compared to the K22 scheme. Less ice crystals are formed in the LP05 experiments with the existence of pre-existing ice crystals.



**Figure 4. Annual grid-mean $N_i$ from 6-year climatology simulations at 250 hPa. The first row shows $N_i$ from the LP05_OGW-Climo and K22_OGW-Climo experiments. The second row shows the differences in $N_i$ between OGW and no_DET experiments (OGW – no_DET) for both the LP05 and K22 schemes, highlighting the contribution from cirrus clouds associated to convective detrainment. The third row presents the $N_i$ differences between OGW and no_OGW experiments (OGW – no_OGW) for both schemes, indicating the presence of orographic cirrus. The fourth row presents the $N_i$ differences between OGW and no_TKE experiments (OGW – no_TKE) for both schemes, reflecting cirrus clouds formed due to turbulence. Dashed lines represent the annual mean -40°C isothermal line, while solid lines indicate the tropopause in the corresponding simulations.**



Fig. 4 shows a global longitude-latitude distribution of annual mean $N_i$ at 250 hPa. Consistent with the results shown in
Fig. 2, the K22_OGW-Climo experiment tends to produce higher ice number concentrations in all three types of simulated
cirrus compared to the LP05_OGW-Climo experiment. In both schemes, cirrus clouds related to convective detrainment
frequently occur over land in low and mid-latitudes. Cirrus clouds due to OGWs mostly occur over mountains and highlands
in mid- and high latitudes. In addition, cirrus clouds due to turbulence have wide-spread contributions globally.

Notably, the K22 scheme distributes high $N_i$ values (>100 L$^{-1}$) more broadly than the LP05 scheme, particularly in mid-
and high latitudes. The extensive regions with high $N_i$ in the K22_OGW_Climo experiment closely resemble the distribution
of cirrus clouds induced by OGWs. This indicates that orographic cirrus clouds are primarily responsible for this shift
between the K22 and LP05 schemes. This shift result in a higher cloud frequency in the K22 scheme compared to the LP05
scheme (Fig. S1). As discussed in Section 2.2, the competition between homogeneous nucleation and pre-existing ice is less
competitive in the K22 scheme than in the LP05 scheme. This reduced competition leads to increased ice crystal formation,
consequently resulting in higher cloud frequency. A further detailed analysis will be presented in the upcoming discussion of
the SPARTICUS campaign.

The difference in ice number between the K22_OGW-Climo and LP05_OGW-Climo experiments is indirectly
influenced by changes in temperature and subgrid-scale vertical velocity variance for ice nucleation. Cirrus clouds warm the
atmosphere below and cool the atmosphere above the cloud layers. In high latitudes, the higher occurrence of cirrus clouds
in the upper troposphere results in warming of the lower troposphere and cooling of the stratosphere (Fig. S2). However, the
overall indirect impact from temperature changes on ice number is not significant in the troposphere, as the temperature
changes are generally small (mostly less than ±0.25 °C).

Changes in the tropospheric temperature also impact global circulation, which in turn affects the subgrid-scale vertical
velocity variance for ice nucleation, as shown in Fig. S3. In the K22_OGW-Climo experiment, vertical velocity variances
increase over mid- and high latitudes in the upper troposphere and lower stratosphere compared to the LP05_OGW-Climo
experiment. These enhanced dynamic factors can lead to a higher number of ice crystals through the occurrence of
homogeneous nucleation. However, the overall indirect impact from changes in vertical velocity on ice number is small in

the troposphere as the changes in vertical velocity are generally small (mostly less than ±0.002 m s⁻¹) , except in the mid-latitudes of the Northern Hemisphere.

In the K22 ice nucleation scheme, the difference in heterogeneous nucleation can be attributed to several other factors beyond changes in vertical velocity variances. These include the parameters used for the activated INP fraction, $\Phi$, and the dust number concentration in simulations. Both the K22 and LP05 schemes account for the activation of coarse mode dust particles for heterogeneous nucleation.

In the K22 experiments, simulated dust aerosol number concentrations tend to be higher compared to those in the LP05 experiment, as depicted in Fig. S4. This difference may stem from changes in the general circulation due to the application of different nucleation schemes. Changes in surface winds may influence dust emissions, while changes in wind fields at different altitudes can influence dust transport and deposition. The increased simulated dust aerosol number concentration in the K22 scheme results in the more ice crystals nucleated heterogeneously compared to the LP05 scheme (Fig. 3c and 3d).

In CAM6, there is an identified bias of aerosols in the upper troposphere related to the wet removal of aerosols by convections, as highlighted by Shan et al. (2021). Fig. S5 and S6 illustrate the $N_i$ and mass-weighted number concentrations of coarse mode dust in the K22_OGW_Shan experiment. With improved aerosol wet removal by convections, the dust aerosol concentration in the K22_OGW_Shan experiment has been reduced (Fig. S6). As a result, ice crystals due to heterogeneous nucleation has decreased, leading to less suppression of homogeneous nucleation. This enhances the homogeneous nucleation, and thus leads to an increase in $N_i$ (Fig. S5). The improved aerosol wet removal by convection based on Shan et al. (2021) appears to effectively optimize the dust number concentrations under the K22 configuration.

**4.2 SPARTICUS Experiments**

Fig. 5a presents the simulated $N_i$ in orographic cirrus during the SPARTICUS campaign for both the LP05_OGW-SP and K22_OGW-SP experiments. Together with simulated microphysical properties (IWC and $D_{num}$) (Fig. S7), both experiments produce results that roughly agree with observational data. However, the simulated IWC and $N_i$ in the K22_OGW-SP experiment tend to be larger than those in the LP05_OGW-SP experiment. Fig. 5b shows the differences of simulated $N_i$ between the reference experiments (OGW) and sensitivity experiments (no_OGW, no_DET and no_TKE). Larger differences in simulated $N_i$ between sensitivity experiments and the reference experiments indicate a more significant




contribution from a respective ice crystal source (OGW-induced, detrained, or turbulence-induced). Specifically, increase or

decrease of microphysical properties in the sensitivity experiments compared to the reference experiments reveals how each

310 source contributes to or inhibits the overall ice number concentrations.

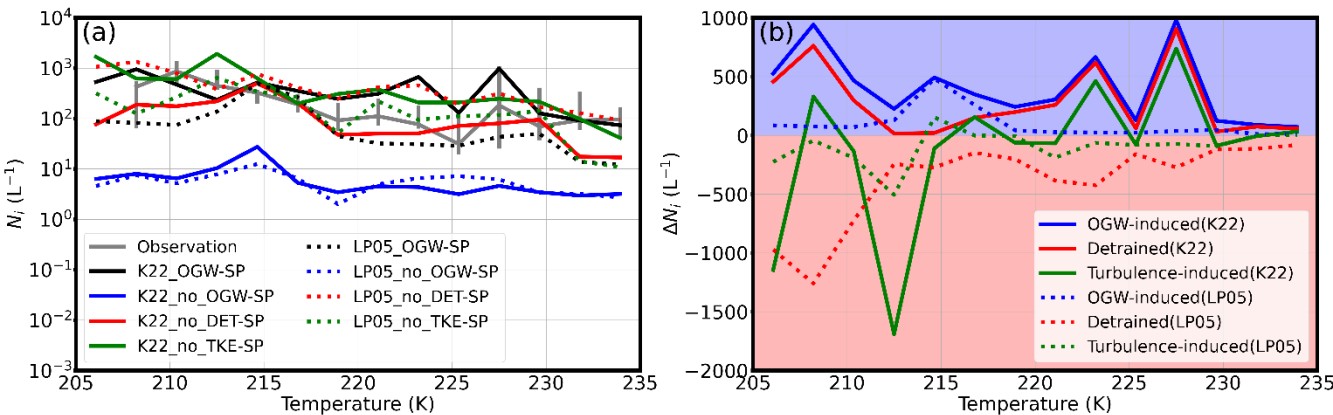

315

**Figure 5. (a) Comparison of $N_i$ between observations and experiments and (b) differences in median $N_i$ values ($\Delta N_i$) between sensitivity tests (no_OGW, no_DET and no_TKE) and reference experiments (OGW) in LP05 and K22 schemes during the SPARTICUS campaign. In panel (a), solid lines represent median $N_i$ values from K22 experiments, while dotted lines represent those from LP05 experiments. The bars indicate observed $N_i$ values,**
320 **ranging from the 25th percentile to the 75th percentile. In panel (b), the number of ice crystals due to OGW is calculated as $N_i$ in OGW experiments minus $N_i$ in no_OGW experiments. The number of ice crystals from convection detrainment is calculated as $N_i$ in OGW experiments minus $N_i$ in no_DET experiments. The number of ice crystals due to turbulence is calculated as $N_i$ in OGW experiments minus $N_i$ in no_TKE experiments. The blue shaded region indicates that the ice crystal source contributes to $N_i$ and increases $N_i$ in the reference experiments. The red shaded**
325 **region indicates that the ice crystal source competes with other sources and inhibits $N_i$ in the reference experiments.**

Fig. 5b demonstrates that in both LP05 and K22 schemes, the changes in $N_i$ ($\Delta N_i$) due to OGWs are always positive.

This indicates that OGWs make significant contributions to the formation of ice crystals in cirrus clouds identified as

orographic cirrus during the observed five-days period. Particularly in regions with temperatures below 215 K, where both

330 schemes simulate their highest $N_i$ peaks, $\Delta N_i$ also peaks positively at the corresponding temperatures. This suggests that

OGW-induced ice crystals are significant contributors to the ice crystals in these cirrus clouds.

The effects of the other two sources are different between the two schemes. In the LP05 scheme, ice crystals from both detrainment and turbulence tend to have inhibition effects on $N_i$. In contrast, the K22 scheme shows clear contributions from detrained ice crystals and varied effects (both contribution and inhibition) from turbulence-induced ice crystals. These differences arise from the distinct representations of competing ice sources between the two schemes.

However, in the high-temperature regions (T>220 K) shown in Fig. 5b, differences in $\Delta N_i$ between the LP05 and K22 schemes are more pronounced. In the LP05 scheme, $\Delta N_i$ due to turbulence and detrainment is generally negative, indicating inhibition effects on $N_i$. On the contrary, the K22 scheme simulates positive $\Delta N_i$ values from these two ice sources, suggesting their contributing effects.

Regarding the simulated $D_{num}$ in the LP05 and K22 experiments (Fig. S8 and S9), the no_OGW experiments generate the largest $D_{num}$ in the experiments. This implies that ice crystals nucleated due to OGW tend to have the smallest $D_{num}$ in the simulations. On the contrary, the smallest $D_{num}$ in the LP05_no_DET-SP experiment indicates that detrained ice crystals contributed the most to the increase in ice crystal sizes. This is consistent with the fixed diameters of 50 μm for detrained ice crystals in the simulations. However, the K22_no_DET-SP experiment does not show similar changes in $D_{num}$ as simulated in the LP05_no_DET-SP experiment. In the LP05 scheme, ice crystals from the detrainment compete for water vapor with the OGW-induced small ice crystals, which is the dominant ice source. The larger detrained ice crystals thus have more opportunity to manifest their size characteristics, increasing the overall $D_{num}$ in the LP05_OGW-SP experiment. This competitive effect is evident when comparing the LP05_OGW-SP experiment with the sensitivity experiment without detrainment (LP05_no_DET-SP), which shows a smaller $D_{num}$, highlighting the dominance of small, nucleated ice crystals from OGWs.

In contrast, in the K22 scheme, the detrained ice crystals do not face significant competition. Consequently, the "large-size" characteristics of these detrained ice crystals are less apparent between the reference experiment (K22_OGW-SP) and sensitivity experiment (K22_no_DET-SP).

Fig. S7 reveals that the K22_OGW-SP experiment generates smaller number-weighted diameters ($D_{num}$) than the LP05_OGW-SP experiment. The combination of smaller $D_{num}$ and larger $N_i$ in the K22_OGW-SP experiment suggests that the K22 scheme tends to produce cirrus clouds with a higher number of smaller ice crystals. This characteristic helps explain



why the K22 scheme results in increased cloud frequency compared to the LP05 scheme. The above analysis of $D_{num}$ highlights the importance of properly parameterizing detrained ice size and competition of ice for water vapor between different sources. Inappropriate settings for detrained ice size can negatively influence the microphysical properties of cirrus clouds. This is especially crucial in models where competition between detrained ice crystals and nucleated ice crystals affects the overall cloud characteristics. Ensuring that these parameters are appropriately represented is essential to reproduce observed characteristics of cirrus clouds microphysics and associated climate effects.

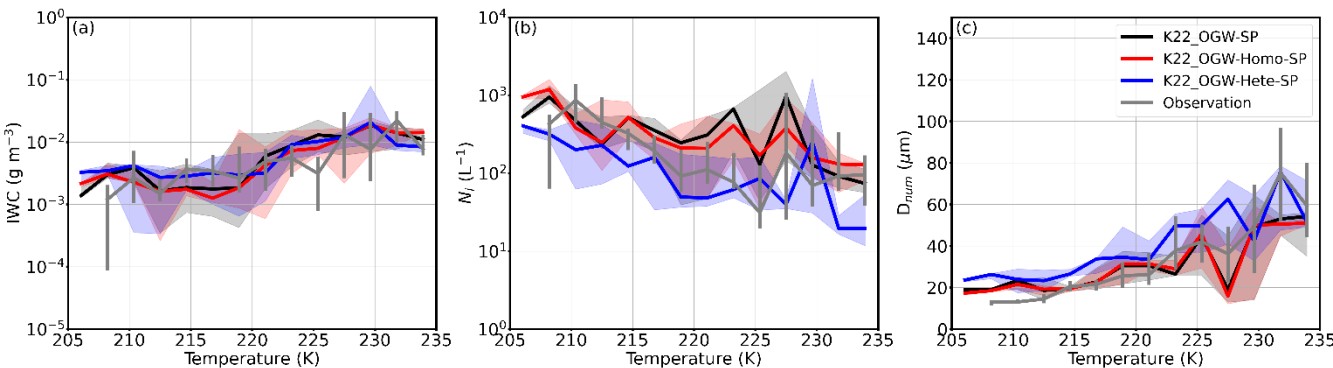

**Figure 6. Comparison of IWC (a), $N_i$ (b) and $D_{num}$ (c) with respect to temperature between observations and K22 sensitivity experiments (K22_OGW, K22_OGW-Homo-SP and K22_OGW-Hete-SP) for orographic cirrus (5 days) during the SPARTICUS campaign.**

A detailed analysis of sensitivity tests with the K22 scheme for the simulations of orographic cirrus clouds has been conducted. As depicted in Fig. 6, the microphysical properties (IWC, $N_i$ and $D_{num}$) in the K22_OGW-SP experiment closely align with those in the K22_OGW-Homo-SP experiment. This alignment suggests that homogeneous nucleation is the dominant mechanism for simulating orographic cirrus during the SPARTICUS campaign using the K22 scheme. This finding is consistent with the results of Lyu et al. (2023) using the LP05 scheme, who also identified the homogeneous nucleation as the dominant mechanism for ice nucleation in orographic cirrus during the SPARTICUS campaign.

## 4.3 ORCAS Experiments

In Region 1, the median values of both simulated and observed IWC are typically low at $10^{-3}$ g m$^{-3}$, implying that less water vapor is available for ice formation in continental environments. As shown in Fig. 7, the median values of simulated $N_i$



generally hover around 3 L$^{-1}$, which is close to the upper limit of observed $N_i$. Simulated $N_i$ are often overestimated, except around 225 K.


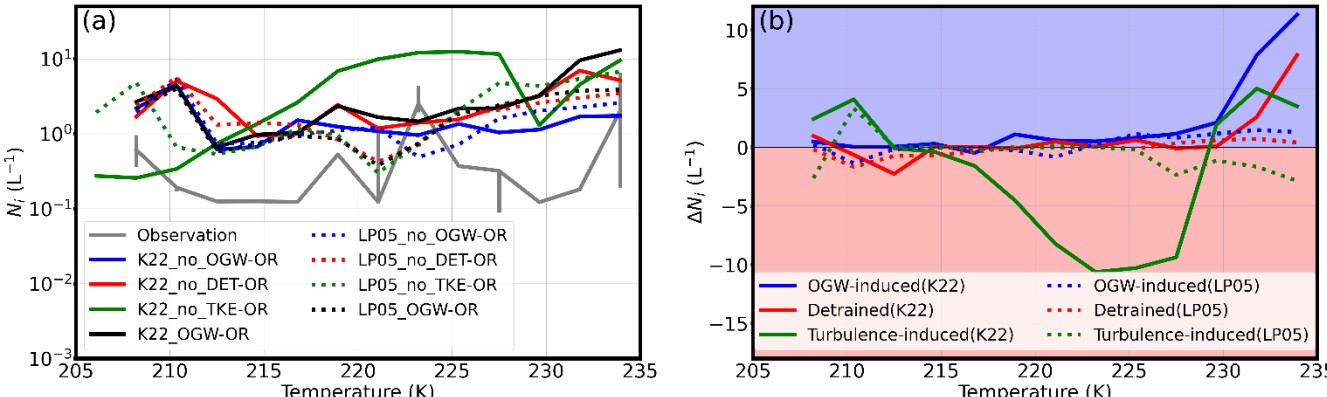

**Figure 7. Same as Figure 5 but for cirrus clouds during the ORCAS campaign in Region 1.**

Regarding observed $N_i$, as shown in Fig. 7, cirrus clouds exhibit multilayer structures with distinct ice sources. There are

pronounced peaks in simulated $N_i$ near 210 K and high $N_i$ values at temperatures above 225 K. At the 210 K level, simulated ice crystals due to turbulence are the primary contributors in both schemes. However, in the LP05 scheme, ice crystals due to OGWs and detrainment tend to inhibit the formation of simulated $N_i$, whereas their effects are minimal in the K22 scheme.

At the lower levels, where high $N_i$ values are observed at temperatures above 225 K, simulated ice crystals due to OGWs and detrainment contribute the most to simulated $N_i$ in both the LP05 and K22 schemes. Ice crystals from turbulence

exhibit inhibiting effects on simulated $N_i$ in the LP05 scheme, while in the K22 scheme, they show varying effects, with inhibition between 215-230 K and contribution to simulated $N_i$ at temperatures ≥ 235 K.

Region 2, located downwind of the southern end of South America and the Antarctic peninsula, features narrow landmass extending into the sea. These high lands create unique conditions for cirrus clouds, characterized by high velocities and relatively high levels of water vapor. The observed median IWC values in Region 2 stay close to 10$^{-2}$ g m$^{-3}$, indicating a

moist environment over the oceans.





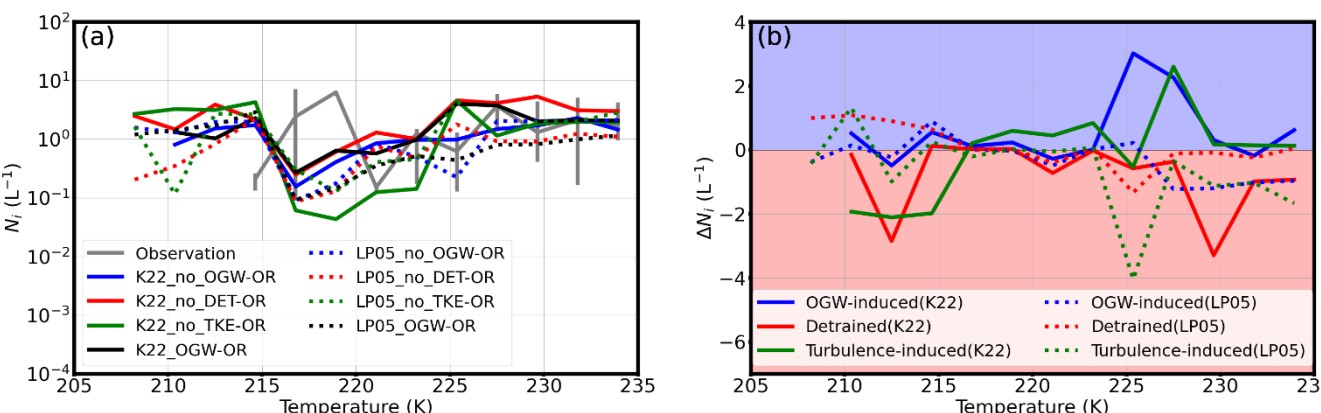

**Figure 8. Same as Figure 7 except in Region 2.**

In Fig. 8a, both observed and simulated $N_i$ show that all experiments (OGW, no_DET, no_OGW and no_TKE) successfully capture the high $N_i$ peaks with multilayer structures. For the high $N_i$ peaks near 215 K, the OGW experiments produce the highest peaks, closely matching the observed magnitude. Fig. 8b suggests that ice crystals contributing to these peaks are predominantly generated from OGWs by mountains and high plateaus in both the LP05 and K22 schemes. The contribution from other sources differs between the schemes: the LP05 scheme shows a preference for contributions from

turbulence and detrainment, whereas the K22 scheme tends to display the inhibition effect. This suggests that simulated $N_i$ peaks around 215 K are related to the mountainous terrain upwind of Region 2.

Notably, while the simulated $N_i$ peaks are around 215 K, the observed $N_i$ peak occurs around 219 K. This bias may be due to an underestimation of ice crystal fall speeds in the model, which could result from either the slow growth of simulated ice crystals or biases in the parameterization of ice crystal fall speed. The stronger competition in the K22 scheme is due to

the wider spread of ice crystals. In the LP05 scheme, the OGW-induced ice crystals tend to remain concentrated over mountains, as shown in Fig. 4. This results in a more localized effect. In the K22 scheme, however, the wider spread of ice crystals allows for competition among different ice sources even at greater distances from mountains.

In the lower part of cirrus clouds below the 225 K level, all three ice crystal sources show competition in the LP05 scheme. In contrast, the K22 scheme shows less competition among these sources. In the K22 scheme, ice crystals due to

OGWs and turbulence contribute positively to simulated $N_i$, and only detrained ice crystals exhibit inhibition effects. Both schemes fail to simulate the same dominant ice source, indicating the absence of dominant ice sources in these simulations.





Previous studies have shown that frontal waves are important dynamic factors for cirrus formation over oceans, while crucial INPs include dust, metallic particles, soot and biological materials (Fan et al., 2016; Froyd et al., 2022; Heymsfield et al., 2017; Kärcher & Ström, 2003; Knopf & Alpert, 2023). In CAM6, however, frontal gravity waves are not included in ice

nucleation processes, and only coarse mode dust is considered as INPs. Future studies are necessary to incorporate these missing ice sources into the model to improve simulations of cirrus clouds over oceans.

In Region 3, the observed IWC median values are even higher than those in Region 2, with maximum values reaching up to $10^{-1}$ g m$^{-3}$. This suggests a water vapor-rich environment for cirrus clouds in this region. In addition, the observed $N_i$ displays a multilayer structure of cirrus clouds (Fig. 9).


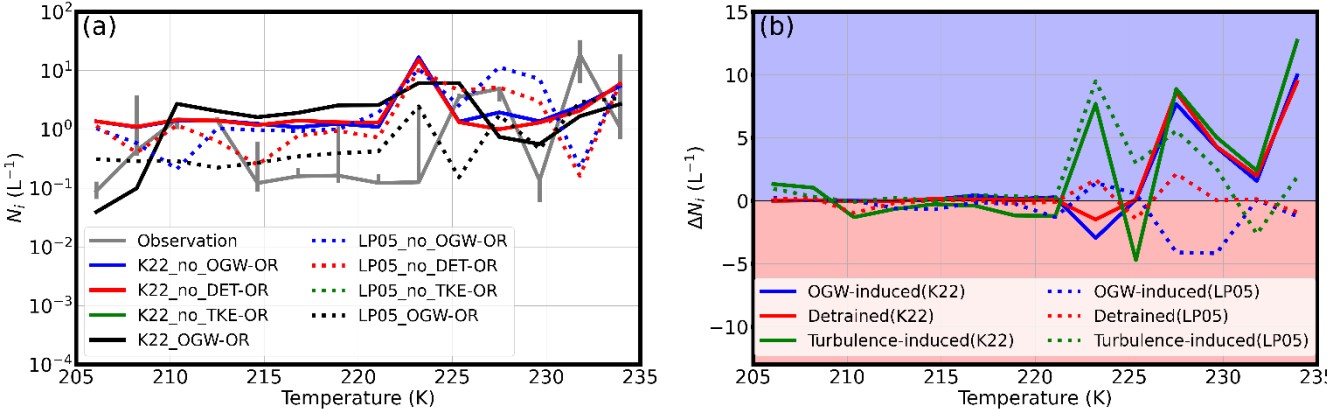

**Figure 9. Same as Figure 7 except in Region 3.**

In higher-level cirrus clouds (T<220 K), both simulated and observed $N_i$ median values are small, usually less than 1 L$^{-1}$.

This indicates weak vertical wind speeds in the oceanic environment. At the cloud top, ice crystals due to turbulence make the most contributions to the simulated $N_i$ peaks when T < 210 K in both schemes (Fig. 9b). However, the simulated $N_i$ in both schemes shows poor agreement with observations, suggesting that turbulence from CLUBB-TKE struggles to simulate the realistic dynamic factors necessary for realistic ice nucleation, such as gravity waves (Gasparini et al., 2023; Kärcher & Podglajen, 2019). The absence of actual ice sources in the simulation points to potentially missing dynamic factors, such as

frontal gravity waves or convective gravity waves, which are likely responsible for ice nucleation under these conditions.

Most of high simulated $N_i$ values are in the lower levels of cirrus (T > 223 K). Both schemes simulate strong contributions from turbulence-induced ice crystals, but they differ in the effects of ice sources due to OGWs and detrainment. The K22 scheme simulates a greater contribution from ice crystals due to OGWs and detrainment, while the LP05 scheme provides more varied effects of these ice sources.

Numerous studies have demonstrated that turbulence from CLUBB-TKE can hardly predict perturbations from gravity waves (Golaz et al., 2002a, 2002b; Huang et al., 2020). Consequently, ice crystals generated by turbulence and from convective detrainment fail to account for critical ice sources over oceans, especially those associated with gravity waves such as frontal and convective gravity waves. To accurately simulate cirrus clouds over oceans, it is necessary to incorporate representations of these key dynamic factors driving ice nucleation.

To improve the model's performance in simulating cirrus microphysical properties over oceans, it is also important to incorporate key INPs into ice nucleation schemes. Previous studies have shown that heterogeneous nucleation plays a significant role over oceans, with mineral dust and metallic particles acting as important INPs (Cziczo et al., 2013; Froyd et al., 2022). Furthermore, soot and biological materials have also been identified (Fan et al., 2016). Including these aerosols may improve the simulation of cirrus clouds by the model.

**4.4  Implication of different behaviours in ice sources with the two nucleation schemes**

Both K22 and LP05 schemes can effectively simulate the dominant ice sources, but they show different effects from the minor ice sources on simulated high $N_i$. For instance, in orographic cirrus, both schemes simulate OGW-induced ice crystals as the dominant contributors, while detrained and turbulence-induced ice crystals show varying effects as minor ice sources. This distinction is useful to identify cirrus types observed during the flight campaigns. Since both schemes can simulate

orographic cirrus clouds, we use these clouds observed during the SPARTICUS campaign as a case study to test this method.

The flight dates when OGW-induce ice sources dominate in the simulations and the simulated $N_i$ aligns closely with observations include 16 days: January 26, 27, February 10, 17, 19, 20, March 14, 17, 19, 30, April 1, 11, 12, 19, 28, and 29. Our method successfully identifies the orographic cirrus observed by Muhlbauer, et al. (2014). Orographic cirrus, referred to as ridge-crest cirrus in their study, are identified on March 19, 30, April 1, 28 and 29. During these days, OGW-induced ice

crystals are the dominant ice sources. Furthermore, we expanded our identification to include 16 days of flight, compared to



the 5 days in Muhlbauer, et al. (2014). This extension increases the number of available data points from 6236 to 15454, thereby making the results more convincing.

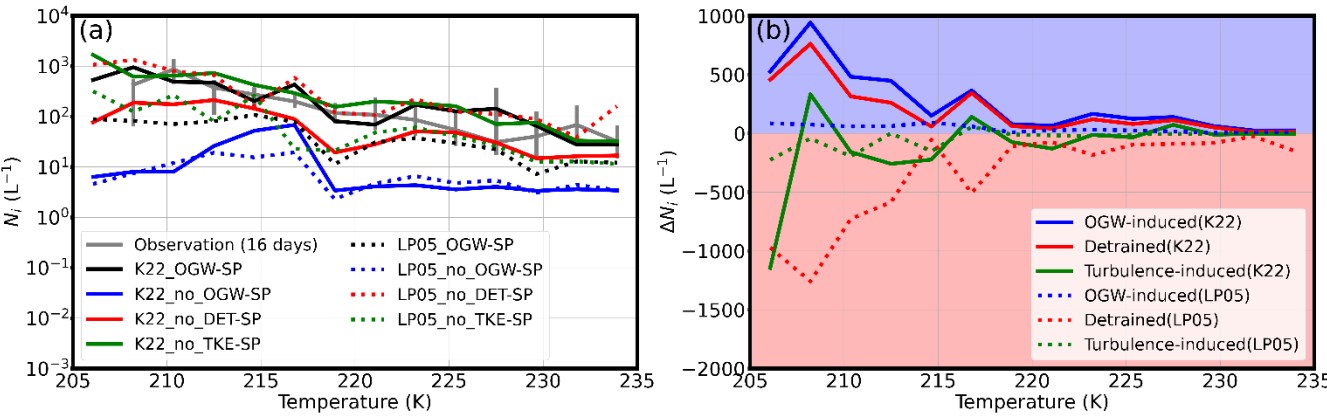

**Figure 10. Same as Figure 5 except for identified orographic cirrus by our approach (16 days of flights).**

Fig. 10 illustrates the microphysical properties of identified orographic cirrus over the 16-day period using our approach. The K22 scheme shows good agreement with observations for these cirrus clouds and the LP05 scheme produces reasonable results within or near the observed range. OGW-induced ice crystals are the dominant contributors in these 16 days of cirrus clouds (Fig. 10b). This indicates that our method is effective and provides a reliable method to distinguish orographic cirrus in flight campaigns.

However, the simulated $N_i$ in the LP05_OGW-SP experiment for 16-day orographic cirrus is underestimated compared to the observations. This discrepancy is largely attributed to the inappropriate size settings for detrained ice crystals, which lead to a higher number of simulated detrained ice crystals over these 16 days. As we discussed in Section 4.2, the stronger competition in the LP05 scheme causes the simulated microphysical properties of cirrus clouds to be more sensitive to biases in other ice sources. Correcting these biases in the simulation would resolve the underestimations of $N_i$ in the LP05 scheme.

## 5.    Summary and Conclusions

This study compares the newly introduced K22 ice nucleation scheme with the default LP05 ice nucleation scheme in the NCAR CAM6 model. The K22 scheme accounts for homogeneous nucleation, heterogeneous nucleation, their
interactions, and competition with pre-existing ice. To investigate sources of ice crystals in cirrus clouds, we conduct 6-year climatology simulations, considering the effects of OGWs on ice nucleation. Additionally, nudged experiments are performed for the SPARTICUS and ORCAS flight campaigns to further compares the two ice nucleation schemes. In all simulations, coarse mode dust is considered as the sole INPs.

In the 6-year climatology experiments, the K22_OGW-Climo experiment shows an increase in grid-mean $N_i$ compared
to the LP05_OGW-Climo experiment. Ice crystals detrained from convection are concentrated in low and mid-latitudes, while those formed due to OGWs are concentrated in mid- and high latitudes. Ice crystals due to turbulence are concentrated in low and mid-latitudes. Notably, homogeneous nucleation plays an important role in the global contribution to the total number of nucleated ice crystals.

The increase in homogeneously nucleated ice numbers in the K22 scheme compared to the LP05 scheme can be
attributed to both direct and indirect reasons. The direct reason relates to differences in the competition parameterization between the K22 and LP05 schemes. To calculate the number of homogeneously nucleated ice crystals, the K22 scheme uses quenching speeds derived from the reduction of supersaturation caused by pre-existing ice crystals. In contrast, the LP05 scheme compares the number of pre-existing ice crystals to the theoretical number of homogeneously nucleated ice crystals in the absence of pre-existing ice. This results in more frequent competition in the LP05 scheme, suppressing homogeneous
nucleation.

The indirect reason arises from the increase in ice number concentrations within the K22 scheme, which leads to higher cloud frequency. This is because smaller ice crystals in the K22 scheme have longer lifetimes, allowing them to travel over broader regions. Changes in cloud frequency induce changes in global temperature, which in turn affect global circulation. Altered circulation dynamics influence the subgrid-scale vertical velocity variance associated with ice nucleation, thereby
impacting ice formation.

In addition, the global increase in coarse mode dust concentrations leads to a higher number of heterogeneously nucleated ice crystals. Improved aerosol wet removal parameterization due to convection can mitigate this by reducing the concentration of coarse mode dust in the upper troposphere.

The nudged experiments conducted during the SPARTICUS flight campaign specifically focus on orographic cirrus clouds. The K22_OGW-SP experiment generates microphysical properties comparable to those of the LP05_OGW-SP experiment, aligning well with observational data. However, it tends to produce a higher number of smaller ice crystals compared to the LP05_OGW-SP experiment. Both the LP05 and K22 schemes identify OGWs as the dominant ice crystal source, but the LP05 scheme exhibits greater competition from detrainment and turbulence sources than the K22 scheme. In addition, the K22_OGW-SP experiment simulates homogeneous nucleation as the dominant mechanism in orographic cirrus formation.

The ORCAS flight campaign is used to further evaluate the simulation results for both the K22 and LP05 schemes. Due to instrument limitations in measuring ice crystals, 2D-C probes are utilized during the ORCAS campaign, allowing for reliable observations of large-size ice crystal microphysical properties ($D_{num} \geq 62.5\mu m$). To better evaluate the results, three regions are divided. Region 1 encompasses flights over high mountains, while Regions 2 and 3 cover flights mostly over oceans, greatly influenced by moist conditions. Region 2, located downwind of the Andes Mountains and high plateaus in Antarctic, is also affected by orographic cirrus clouds, which impact the observed cloud microphysical properties.

The overall microphysical properties in these three regions are not well represented in the OGW experiments for both the K22 and LP05 schemes compared to observations. This discrepancy arises because the model struggles to simulate the dominant ice sources over oceanic regions. It underscores the importance of incorporating the dominant ice sources into the model to effectively reproduce the observed characteristics of cirrus clouds.

The differences in moisture availability and dynamic environmental conditions between land and ocean result in distinct cloud microphysical behaviours and ice nucleation processes, resulting in unique characteristics of cirrus clouds in these two environments.

Over land, particularly in mountainous regions, the air is relatively dry, leading to low observed and simulated IWC. However, the high vertical velocities provided by mountains foster favourable conditions for homogeneous nucleation,



which often becomes the dominant nucleation mechanism in orographic cirrus clouds. These clouds can exhibit high ice number concentrations and reach very high altitudes. Consequently, incorporating OGWs in ice nucleation processes allows the model to better simulate cirrus clouds over land.

In contrast, over oceans, the atmosphere is rich in water vapor, resulting in high observed IWC. However, the lack of vertical velocity sources in the upper troposphere over oceans results in heterogeneous nucleation being the dominant mechanism. Ice crystals due to gravity waves, such as frontal and convective gravity waves, can be important ice sources in these regions. In addition, some critical INPs are absent in current ice nucleation schemes. As a result, the model performs poorly in simulating cirrus clouds over oceans. To address the absence of crucial ice sources in these simulations, further studies should incorporate additional dynamic factors, such as frontal gravity waves and convective gravity waves, as well as other types of INPs.

In conclusion, an accurate representation of the competition mechanism in the ice nucleation scheme is crucial, particularly for the interactions between pre-existing and newly nucleated ice crystals, as these interactions can significantly influence the simulated microphysical properties of cirrus clouds. While many studies primarily focus on the dominant ice sources, minor sources are often overlooked. However, our studies suggest that minor ice sources may also influence cloud microphysical properties through competition mechanisms between pre-existing and newly nucleated ice crystals. Understanding these dynamics is essential for accurately simulating the behaviours and characteristics of cirrus clouds under different atmospheric conditions.

Moreover, distinguishing ice crystal sources has long posed a significant challenge in the study of cirrus clouds. The different behaviours between dominant and minor ice sources in high $N_i$ regions with the K22 and LP05 schemes provide a reasonable method for distinguishing cirrus types in observations, especially for orographic cirrus. By applying this method to categorize orographic cirrus in the SPARTICUS campaign, we identify 16 days of flight during which OGW-induced ice source dominates the ice formation, with no significant underestimation of $N_i$ in either scheme. These identified flights exhibit a good agreement in microphysical properties with observations, proving that this method is effective for distinguishing orographic cirrus from observations.



Furthermore, our comparison of simulated cirrus clouds with observations highlights the need for refining the model representation of processes governing cirrus cloud evolution, including detrainment, ice crystal growth and ice crystal sedimentation.

**Code and data availability.** For readers interested in replicating specific aspects of our study, we encourage them to
contact the corresponding authors of the cited papers for access to the underlying code and data.

**Author contributions.** KL: incorporated K22 scheme into CAM6, conducted simulations, analyzed results, wrote the article; XL: provided guidance, reviewed the manuscript; BK: provided K22 nucleation parameterization and reviewed the manuscript.

**Competing interests.** At least one of the (co-)authors is a member of the editorial board of Atmospheric Chemistry of Physics.

**Disclaimer.** Publisher's note: Copernicus Publications remains neutral with regard to jurisdictional claims made in the text,
published maps, institutional affiliations, or any other geographical representation in this paper. While Copernicus Publications makes every effort to include appropriate place names, the final responsibility lies with the authors.

**Acknowledgement.** This work was supported by the National Aeronautics and Space Administration (NASA) grant (No. ROSES-2020 80NSSC21K1457).

**Financial support.** This research has been supported by the National Aeronautics and Space Administration (NASA) grant (No. ROSES-2020 80NSSC21K1457).

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
