# Peer review of "Exploring Sources of Ice Crystals in Cirrus Clouds: Comparative Analysis of Two Ice Nucleation Schemes in CAM6"

_EGUsphere, 2024_

## Referee Comment (RC2)

Review of "Exploring Sources of Ice Crystals in Cirrus Clouds: Comparative Analysis of Two Ice Nucleation Schemes in CAM6" by Lyu et al. [Research Article, egusphere-2024-4144]

This study coupled a novel ice nucleation parameterization scheme based on Karcher (2022) into CAM6 and compared its representation of cirrus ice cloud microphysics against the default Liu and Penner (2005) scheme, with a particular focus on ice sources in cirrus cloud formation. The authors conducted a thorough assessment using both long-term simulations and case studies from the SPARTICUS and ORCAS campaigns. Their findings revealed several similarities between the two schemes, such as the climatological location of orographic gravity wave (OGW)-induced ice crystals and the same dominant source (OGW-induced) for orographic cirrus. Notable differences were also identified, primarily attributed to the distinct nucleation/competition mechanisms within the two schemes. Overall, the manuscript is well written, but its structure could be improved for better readability. For example, the excessive use of short paragraphs disrupts the flow, and combining some of them could enhance clarity. This work holds significant potential for advancing ice cloud simulations, particularly in refining parameter tuning and improving the representation of competition mechanisms. However, my major concern is the lack of sufficient physical explanations and robust evidence for the model biases and the differences found between the two schemes. If these issues can be addressed, I believe this paper will be well-suited for publication in ACP.

**Major comments:**

1. A key concept in this study is the competition between homogeneous and heterogeneous freezing in ice cloud formation. The authors argued that the competition is stronger in the LP05 scheme than in K22 due to differences in their parameterization of homogeneous nucleation occurrence. However, this claim appears to be more of an assumption than a rigorously validated conclusion, as it is not directly substantiated from the parameterization formulas (not shown by the authors). The authors have used this assumption multiple times (e.g., Lines 246-248 and 268-270) to explain discrepancies in simulated ice cloud microphysics between the two schemes. I think a more appropriate way would be to first make this assumption explicitly and then examine it using supporting evidence from simulation results.

The authors found that fewer new ice crystals form in LP05 with the presence of pre-existing ice crystals (Line 247), which aligns with the assumption of stronger competition in LP05. However, a critical underlying assumption is that both schemes should have a similar or comparable number concentration of pre-existing ice crystals. If the LP05 experiments contain a higher concentration of pre-existing ice crystals than K22, it becomes difficult to determine whether the reduction in new ice formation is genuinely due to stronger competition in LP05 or a result of differing initial conditions. To address this issue, the authors should ensure that the number concentration of pre-existing ice crystals is close across experiments or, at the very least, discuss the potential influence of variations in pre-existing ice concentrations on their results.

Additionally, the proposed indirect explanation for the increase in ice number concentration in K22 is not sufficiently substantiated. For example, the authors did not show evidence on how the

changed circulation dynamics impact the sub-grid turbulence, making this explanation remain speculative rather than a well-supported conclusion.

2. Since one purpose of this paper is to evaluate the K22 scheme, incorporating climatological (6-year) observational data is important for assessing the performance of both schemes. If obtaining global vertical profiles is challenging, bulk or regional observational data would still be valuable in determining whether K22 improves ice cloud simulations compared to LP05 from a climatological perspective.

3. To deepen the insights of this study, the authors could discuss the potential impact of incorporating K22 into CAM6 on high cloud feedback. For example, if the proposed indirect mechanism for the higher ice number concentrations in K22 is true, large-scale circulation changes induced by global warming could modify sub-grid turbulence, subsequently affecting ice nucleation, cloud frequency, and longwave radiative effects.

4. One structural issue in the manuscript is the overuse of short paragraphs, which disrupts the flow of the text. I recommend revisiting the paragraph structure and merging shorter paragraphs with logically related content to enhance readability and coherence. I will provide some specific suggestions in the minor comments, though they are not exhaustive.

**Minor comments:**

L7: I'd suggest reorganizing the abstract into two paragraphs or three at most.

L88: Please give a brief reason why both field campaigns are used for validation. Any differences between these two or just for increasing the sample size?

L98: No definition for "DET" upon its first appearance.

L127: "thermaldynamic" to "thermodynamic"

L150: Suggest moving this paragraph up.

L152: "compared" to "compared to"

L211: Since OGW-induced cloud nucleation is a very important source for ice cloud formation, I'd suggest comparing the climatology simulation results between land and oceans. The results over the land might be more contrasting between the two schemes.

L215: Why more cirrus due to OGWs in high latitudes, particularly near the Poles (Figures 2e and 2f)?

L216: Please clarify the physical mechanisms for turbulence-induced ice nucleation.

L232: "results" to "resulting", "resemble" to "resembles"

L279: Any physical explanations for changes in sub-grid turbulence?

L303: "Together with … (Fig. S7)" to "Together with simulated IWC and $D_{num}$ (Fig. S7)"

L316: What does "Temperature in X-axis" represent? Pressure-level mean temperature?

L330: "$\Delta N_i$" to "$\Delta N_i$ due to OGWs"

L336-337: Be specific. It looks dependent on the source types.

L351: As in K22 the detrained ice crystals do not have a significant competition, I'd expect that $D_{num}$ is slightly lower in K22_no_DET-SP (red lines in Fig. S9) than in K22_OGW-SP. However, why is it slightly higher in K22_no_DET-SP when T is less than 227 K?

L388: Suggest moving this paragraph up

L391: Why is its magnitude so large between 220 and 230 K?

L402: It seems like no_TKE experiment shows the highest peak. Please double check.

L413: Please rephrase this sentence.

L456: "OGW-induce" to "OGW-induced"

L456: Are these dates selected when the simulations of both schemes align with the observations?

L481: "considering" to "with a focus on"

L490-495: Are there any physical reasons, or formula-related proofs? If not, the first reason is more like an assumption.

L496-500: Have you examined the changes in large-scale circulations and their association with sub-grid turbulence variations? If not, you would have to soften your tone when proposing the indirect reason.

L524: Please move it to the last paragraph.

L532: Be specific for these critical INPs.

---

## Author Comment (AC1)

We thank the two anonymous reviewers for their constructive comments. Below, we explain how the comments are addressed and make notes of the revisions in the revised manuscript. The reviewers' comments are in blue color. Our replies are in black, and our corresponding revisions in the manuscript are in red (line numbers are based on the tracked version of the revised manuscript).

Recommendation: Return to Authors for Major Revisions

Overview:

The study by Lyu et al. (2024), titled "Exploring Sources of Ice Crystals in Cirrus Clouds: Comparative Analysis of Two Ice Nucleation Schemes in CAM6," is interesting and valuable for the field of cloud physics and development of Earth system model. However, this manuscript needs significant improvements before being published. Below are my comments, questions and suggestions.

Thank you very much for your helpful and constructive comments.

Major comments/questions:

1. The grammar and wording of this manuscript is poor. I added some revision suggestions in the minor comment/questions part, but strongly recommend the authors carefully go through the whole manuscript to improve the writing. Also, the logic in several sections is difficult to follow, which also needs to improve.

Thank you very much for your comments. We have carefully reviewed the manuscript to improve both the writing and the logic flow. First, we refined the language by adding more detailed explanations and clarified previously ambiguous points. Second, we improved the structural coherence of the manuscript by reorganizing paragraphs and sections to ensure a more logical presentation of the content.

2. Generally, the INP activation differences between K22 and LP05 could be explained by activation efficiency difference and aerosol difference. It is great that both factors are analyzed in Section 4.1. However, it seems the biases analysis in Sections 4.2 and 4.3 only focuses on INP activation efficiency, while totally neglects aerosol concentration. This really needs improvement by adding aerosol concentration evaluation and comparison.

Thank you very much for your comments. We did not include aerosol concentration evaluation in Section 4.2 and Section 4.3 because there were no aerosol measurements during the flights. Both the SPARTICUS and ORCAS campaigns primarily focused on cirrus clouds, and no corresponding aerosol observational data were available to support the model validation. However, we agree with the reviewer that comparing the aerosol concentrations between the two schemes adds important value. Therefore, we have included the comparisons of simulated coarse mode dust number concentrations during the SPARTICUS and ORCAS campaigns (Figures R1-1, R1-2, R1-3, and R1-4). These

figures have been included in the supplementary materials, and the corresponding text has been incorporated into Section 4.2 and 4.3 to reflect these additions.

The relevant paragraph has been modified as follows (line 593-596):

"The simulated coarse mode dust number concentrations are shown in Fig. S17, which shows higher values in the K22 scheme than those in the LP05 scheme. However, the dust concentrations are very low ($< 1$ L$^{-1}$) in both schemes, which supports the dominance of homogeneous ice nucleation for cirrus cloud formation during the SPARTICUS campaign."

(line 610-612):

"The simulated coarse mode dust number concentrations are presented in Fig. S19, which shows higher values with the K22 scheme compared to the LP05 scheme."

(line 644-645):

"Figure S20 shows the simulated coarse mode dust number concentrations, with the K22 scheme generally simulating higher dust concentrations compared to the LP05 scheme."

(line 692-694):

"Simulated coarse mode dust number concentrations from both schemes are compared in Fig. S21, showing that the K22 scheme simulates much higher dust concentrations than the LP05 scheme."

[Figure]

Figure R1-1. Comparison of coarse mode dust number concentrations between LP05_OGW-SP and K22_OGW-SP during the SPARTICUS campaign.

[Figure]

Figure R1-2. Comparison of coarse mode dust number concentrations between LP05_OGW-OR and K22_OGW-OR during the ORCAS campaign in Region 1.

[Figure]

Figure R1-3. Comparison of coarse mode dust number concentrations between LP05_OGW-OR and K22_OGW-OR during the ORCAS campaign in Region 2.

[Figure]

Figure R1-4. Comparison of coarse mode dust number concentrations between LP05_OGW-OR and K22_OGW-OR during the ORCAS campaign in Region 3.

3. All conclusions in this study are based on one precondition that Ni is mainly dominated by INP activation, while other source/sink terms like secondary ice production, ice sedimentation and sublimation are totally neglected. In my view, this leads to incomplete discussion. More effort is needed to verify that the INP activation dominates Ni.

Thank you very much for your comments. In this study, we do not have the precondition that $N_i$ is mainly dominated by ice nucleation, and acknowledge that other processes such as secondary ice production, ice sedimentation and sublimation can also impact $N_i$ and contribute to the discrepancies between model results and observations. We have added some discussions.

For the SPARTICUS campaign, the relevant paragraph has been modified as follows (line 602-604):

"Additionally, discrepancies between the simulations and observations may stem from limitations in model representations of other microphysical processes, such as ice depositional growth, cloud ice to snow autoconversion and accretion, and ice sedimentation."

For the ORCAS campaign where multi-layer cirrus clouds frequently occurred, ice sedimentation and sublimation can be important for determining $N_i$ in these clouds. The model may also miss other sources of ice crystals in these clouds when compared with observations. The relevant paragraph has been modified as follows (line 679-680 and line 685-687):

"The fact that no $\Delta N_i$ values from a single source are overall positive in both schemes may suggest that the dominant ice source is missing from the model."

"In addition, other important $N_i$ source and sink processes, such as secondary ice production, ice sublimation and sedimentation should be examined."

In the Summary and Conclusions section, we added (line 871-873):

"Furthermore, our comparison between simulated cirrus clouds with observations highlights the need for refining the model representation of key processes governing cirrus cloud evolution. They include detrainment, ice crystal growth mechanisms (ice deposition, and accretion), secondary ice production, sublimation, and ice crystal sedimentation."

4. The competition between homogeneous and heterogeneous freezing seems to be sensitive to three important physical mechanisms: ice detrainment, OGW and TKE, and their impacts are discussed. However, the homogeneous and heterogeneous freezing rates in experiments are not shown, and the corresponding conclusions are based on speculation. Please show the freezing rates to support your conclusions.

Thank you very much for your comments. The MG2 scheme computes ice number ($N_i$(t) which represents ice number at time t) using both ice number ($N_i$(t-1)) and number tendency ($\Delta N_i$(t-1)) at time t-1 with time step (30 minutes). This allows us to distinguish between pre-existing ($N_i$(t-1)) and newly generated ice crystals ($\Delta N_i$(t)) at a specific time t and model grid. However, the model calculates the number of homogeneously and heterogeneously nucleated ice crystals at each time step and model grid and then derives the corresponding freezing rates.

Fig. R1-5 and Fig. R1-6 show the annual mean ice number tendency due to heterogeneous nucleation ($\Delta N_{i\_het}$) from 6-year climatology simulations, shown as zonally distributed (Fig. R1-5) and at 250 hPa (Fig. R1-6). Both schemes simulate $\Delta N_{i\_het}$ are concentrated at mid- and high-latitudes in the upper troposphere (Fig. R1-5a, b), indicating that heterogeneous nucleation is most active in these regions. High $\Delta N_{i\_het}$ values extend over land and ocean regions (Fig. R1-6a, b). Compared to the LP05 scheme, the K22 scheme simulates higher $\Delta N_{i\_het}$ values in mid and high latitude regions. This enhancement aligns with the higher coarse mode dust number in the K22_OGW-climo experiment (see Fig. S4 in the supplementary material). Both schemes show similar $\Delta N_{i\_het}$ distributions from convective detrainment between no_DET and OGW experiments (Fig. R1-5c, d and Fig. R1-6c, d), indicating that heterogeneous nucleation is not directly influenced by convective detrainment. In contrast, the no_OGWs experiments (Fig. R1-5e, f and Fig. R1-6e, f) show pronounced reduction in $\Delta N_{i\_het}$ in the mid- and high latitudes compared to OGW experiments, revealing the significant role of OGWs in enhancing heterogeneous nucleation. This effect is especially evident in the K22 scheme, which shows substantial $\Delta N_{i\_het}$ reductions over continental regions, especially over mountainous areas such as the Himalayas, Andes, Alps and Rockies, indicating a strong sensitivity of heterogeneous ice nucleation to OGWs. The LP05 scheme exhibits more limited changes in $\Delta N_{i\_het}$, suggesting a weaker enhancement from OGWs. These different results between the two schemes are due to their distinct parameterizations of heterogeneous nucleation. For turbulence-induced $\Delta N_{i\_het}$ (Fig. R1-5

g, h and Fig. R1-6g, h), both the K22_noTKE and LP05_noTKE experiments simulate reduced $\Delta N_{i\_het}$ compared to their respective OGW-Climo experiments. This result indicates that turbulence reinforces INP activation.

Fig. R1-7 and Fig. R1-8 present the zonal mean and 250 hPa ice number tendency due to homogeneous nucleation ($\Delta N_{i\_hom}$). In both schemes, homogeneous nucleation primarily occurs over high mountains in mid- and high latitudes, as well as in the tropical tropopause layers (TTL). Overall, the K22 scheme produces larger $\Delta N_{i\_hom}$ compared to the LP05 scheme. The LP05_no_DET experiment exhibits enhanced $\Delta N_{i\_hom}$ in the tropopause (Fig. R1-7c and R1-8c), compared to the LP05_OGW-Climo experiment, indicating that convective detrainment suppresses homogeneous nucleation in the LP05 scheme. In contrast, the K22_no_DET experiment exhibits limited changes compared to the K22_OGW-Climo experiment, indicating that detrainment has a limited effect on homogeneous nucleation in the K22 scheme (Fig. R1-7d and R1-8d,). Both schemes simulate significantly reduced $\Delta N_{i\_hom}$ over high mountains compared to the OGW experiments (Fig. R1-7e, f and R1-8e, f), emphasizing the role of OGWs in promoting homogeneous nucleation. Similarly, the no_TKE experiments (Fig. R1-7g, h and R1-8g, h) produce reduced $\Delta N_{i\_hom}$ in the TTL for both schemes, revealing that turbulence enhances homogeneous nucleation in this region.

The above discussions have been included in the manuscript (line 437-463).

[Figure]

Figure R1-5. Annual zonal ice number tendency due to heterogeneous nucleation $\Delta N_{i\_het}$ from 6-year Climatology simulations in the upper troposphere (above 600 hPa). Dashed lines indicate the annual mean -40 ℃ isothermal line, and solid lines represent the tropopause in the corresponding simulations.

[Figure]

Figure R1-6. Annual ice number tendency due to heterogeneous nucleation $\Delta N_{i\_het}$ from 6-year climatology simulations at 250 hPa.

[Figure]

Figure R1-7. Annual zonal ice number tendency due to homogeneous nucleation $\Delta N_{i\_hom}$ from 6-year Climatology simulations in the upper troposphere (above 600 hPa). Dashed lines indicate the annual mean -40 °C isothermal line, and solid lines represent the tropopause in the corresponding simulations.

[Figure]

Figure R1-8. Annual ice number tendency due to homogeneous nucleation $\Delta N_{i\_hom}$ from 6-year climatology simulations at 250 hPa.

Minor comments/questions:

1. Line 11: Consider revising "detrained" to "detrainment."

Thank you very much for your comments. We have modified the word as suggested.

The relevant paragraph has been modified as follows (line 10-11):

"To investigate ice formation in cirrus clouds, sensitivity tests are conducted to analyze three ice sources from orographic gravity waves (OGWs), convection detrainment, and turbulence."

2. Lines 15-16: This sentence could be clarified. Do you mean that ice crystals from detrainment and formed by turbulence are primarily concentrated in low- and mid-latitudes?

Thank you very much for your comment. We have modified the relevant sentence as suggested.

The relevant paragraph has been modified as follows (line 15-17):

"Both schemes simulate that convection detrained and turbulence-induced ice crystals are concentrated in low- to mid-latitudes, whereas OGW-induced ice crystals are concentrated in mid- to high latitudes."

3. Lines 19-21: Further clarification is needed. Since Lines 15-16 indicate that the importance of ice sources varies by latitude, could you explain why orographic gravity waves (OGWs) are identified as the dominant ice source?

Thank you very much for your valuable comment. We have included additional explanations regarding orographic cirrus clouds (i.e., cirrus over high terrains). In these clouds, ice crystals are primarily generated by OGWs.

To better address the issue, the relevant paragraph has been modified as follows (line 20-21):

"In orographic cirrus over high terrains at mid- to high latitudes, both schemes identify OGW-induced ice crystals as the dominant ice source."

4. Lines 26-27: Please provide references to support the statement: "These ice clouds can reflect solar radiation back to space, cooling the planet."

Thank you very much for your comment. We have included a reference to support the statement. This is also supported by the reference by Liou (1986).

The sentence has been modified as follows (line 27-28):

"These ice clouds can reflect solar radiation back to space, cooling the planet (Chen et al., 2024; Forster et al., 2023). """

5. Lines 36-41: Citations are needed to substantiate the claim that homogeneous freezing typically results in cirrus clouds with $N_i > 100$ $L^{-1}$, whereas heterogeneous freezing generally produces cirrus with $N_i < 10$ $L^{-1}$.

Thank you very much for your comment. These results are based on Heymsfield et al. (2017) and Froyd et al. (2022). We have added these citations as follows (in line 42-43):

"This process generally produces low ice number concentrations (< 100 $L^{-1}$) (Heymsfield et al., 2017; Froyd et al., 2022)."

6. Line 48: Does "vertical velocity" refer specifically to subgrid vertical velocity? If so, is grid-scale vertical velocity considered in the ice-nucleating particle (INP) scheme?

Thank you very much for your comment. Yes, this refers to subgrid-scale vertical velocity. In current GCMs, the horizontal grid spacing is about 1 degree (~100 km), which allows grid-scale vertical velocity can be represented in the temperature and supersaturation. Because subgrid-scale motions dominate the vertical uplift necessary for ice nucleation, subgrid-scale vertical velocity is explicitly incorporated into the ice nucleation parameterization.

The relevant paragraph has been rearranged and modified as follows (line 50-51):

"For example, most GCMs treat turbulence as the sole subgrid-scale vertical velocity mechanism driving ice nucleation."

7. Line 53: As mentioned in Lines 32-33, cirrus clouds can form through detrainment or in-situ nucleation. Could you clarify which mechanism is responsible for orographic cirrus formation?

Thank you very much for your comment. We have modified the sentence as follows (line 38):

"Ice crystals in in-situ cirrus clouds, such as orographic cirrus over high terrains, are primarily nucleated by aerosols."

8. Lines 57-60: A more detailed explanation of the complexity of INP parameterization would be helpful. Two major sources of uncertainty are (1) aerosol properties and (2) supersaturation levels. This study focuses on refining supersaturation calculations by incorporating OGW effects and evaluating the sensitivity of different INP activation efficiencies (LP05 and K22 schemes). If this understanding is correct, consider refining this section to better reflect this logic.

Thank you very much for your valuable comment. We have modified the corresponding sentences to highlight the two sources of uncertainty from your comment as follows (line 62-70):

"Aerosols such as dust, soot, metallic particles, and biological particles, can act as INPs, inducing heterogeneous nucleation and potentially suppressing homogeneous nucleation (Fan et al., 2016; Froyd et al., 2022; Heymsfield et al., 2017; Kärcher & Ström, 2003; Knopf & Alpert, 2023). The activation efficiency of INPs is determined by their chemical components, which is highly dependent on their sources (Beall et al., 2022; Chen et al., 2024; Tobo et al., 2019). Limited knowledge of the number concentration, chemical composition, and activation efficiency of INPs in the upper troposphere complicates the model prediction of cirrus clouds microphysical properties (Knopf & Alpert, 2023). Moreover, currently conventional GCMs cannot resolve the subgrid-scale vertical

velocity, which drives the water vapor supersaturation for ice nucleation, posing additional uncertainty for model simulations."

9. Lines 80-84: Could you confirm whether subgrid-scale vertical velocities from OGW and turbulence are explicitly incorporated into the INP scheme? Additionally, is the turbulence-driven vertical velocity derived from TKE? If so, please clarify this point in the text.

Thank you very much for your valuable comments. Yes, subgrid-scale vertical velocities from OGW and turbulence are explicitly incorporated into the ice nucleation schemes.

Yes, the turbulence-driven vertical velocity is derived from TKE. We have included additional explanations to clarify this point as follows (line 91-95):

"Since CLUBB effectively represents turbulence with a small Richardson number but struggles to produce perturbations caused by gravity waves (Golaz et al., 2002a, 2002b; Huang et al., 2020), subgrid-scale vertical velocities from orographic gravity waves (OGWs) and turbulence are incorporated into the ice nucleation schemes (Lyu et al., 2023). The turbulence-driven vertical velocity is derived from TKE calculated by CLUBB."

10. Line 85: Should "cloud-borne state" be revised to "ice-borne state"? Are you referring to aerosols incorporated into ice crystals?

Thank you very much for your comment. In CAM6, cloud-borne state is used to represent both ice-borne and liquid-borne aerosols in warm and cold clouds. There is no separation of ice-borne state and liquid-borne state. So, we keep using "cloud-borne" state.

11. Lines 91-95: The LP05 scheme should be described as explicitly as the K22 scheme. Could you provide additional details on how subgrid-scale vertical velocity is used to compute supersaturation? How are aerosol properties (e.g., number concentration, size distribution, and chemical composition) incorporated into INP activation calculations? Additionally, are there differences in how the LP05 scheme treats homogeneous versus heterogeneous freezing, and how does it account for competition between these two processes?

Thank you very much for your comments. We have added further explanations (Section 2.2.2) as you suggested (line 176-194):

"The LP05 ice nucleation scheme incorporates two primary mechanisms: homogeneous and heterogeneous nucleation (Liu & Penner, 2005). It is based on fitted simulation results from a cloud parcel model with varying vertical velocities. The maximum supersaturation is determined in the parcel model from the balance between the production due to adiabatic cooling by updrafts and loss due to vapor deposition on ice crystals. The number of nucleated ice crystals is derived based on ice supersaturation,

temperature, aerosol number concentration and composition, and vertical velocity. Subgrid vertical velocity can be derived from TKE calculated by CLUBB, from OGWs, or from the combined contribution of both components.

Homogeneous nucleation in the LP05 scheme, similar to the K22 scheme, adopts the parameterization by Koop et al. (2000). Sulfate aerosols in the Aitken mode with diameters greater than 0.1 μm is applied to fit to ice number concentrations (Gettelman et al., 2010). On the other hand, heterogeneous nucleation considers the coarse mode dust as potential source of INPs. The number of ice crystals formed due to heterogeneous nucleation $n$ in the LP05 scheme is calculated using $n = n_{dust} \cdot \Phi(T, w, S_i)$, where $n_{dust}$ is the coarse mode dust number concentration from MAM4, and $\Phi$ is active aerosol fraction, empirically derived as a function of temperature ($T$), vertical velocity ($w$), and ice supersaturation ($S_i$).

The LP05 scheme considers the competition between homogeneous and heterogeneous nucleation. It determines the critical dust INP concentration, above which homogeneous nucleation is completely switched off. Below that, homogeneous nucleation occurs partially and is gradually transitioned to the pure homogeneous nucleation at lower INP concentrations. The LP05 scheme is modified to consider the effect of pre-existing ice crystals (Shi et al., 2015), which is parameterized by reducing the vertical velocity for ice nucleation as a result of water vapor deposition on pre-existing ice."

12. Subsections 2.1 and 2.2: The organization of these sections could be improved for readability. Consider the following structure: (1) A brief introduction to CAM6 (e.g., currently covered in Lines 74-83). (2) A detailed explanation of the LP05 and K22 schemes (e.g., Lines 91-95, but needs to be expanded, and content from Subsection 2.2). (3) A description of the experimental setup (e.g., Lines 88-90 and 96-100), with additional details on the objectives and configurations.

Thank you very much for your good suggestions. We have reorganized the paragraphs as you suggested (see Section 2.1, 2.2, 2.3).

13. Line 117: Does Equation (1) apply to homogeneous or heterogeneous freezing? Lines 109-114 discuss homogeneous freezing but do not include equations, whereas Lines 122-123 reference heterogeneous freezing. Clarifying this distinction would be helpful.

Thank you very much for your valuable comments. Equation 1 only applies to heterogeneous nucleation on INPs. We have clarified this.

14. Line 118: To which aerosol species does the INP number concentration here correspond? How does K22 handle aerosol mixing state parameterization?

Thank you very much for your comments. This study considers only coarse mode dust as INPs. We don't consider the difference in the INP activation efficiency depending on the dust mixing state.

15. Subsection 2.2: The mathematical formulation of K22 is unclear. Could you clarify how the loss term in Equation (3), Wq,het, and Wq,pre are computed? Also, does w (the first term on the right-hand side of Equation (5)) represent subgrid vertical velocity derived from TKE and/or OGW?

Thank you very much for your comments. $w_q$ including contributions from heterogeneous nucleation and pre-existing ice is calculated as follows (line 164-167):

"Quenching velocities $w_q$ are defined as:

$$w_q = \frac{\int_0^s \frac{4\pi}{vn_{sat}ds} \frac{dn}{ds} \left( \int_{\tau(s)}^{t(s)} r^2 \frac{dr}{dt} dt \right) ds'}{a(s+1)}, \qquad (4)$$

where the loss term includes contributions from heterogeneous nucleation and pre-existing ice."

Yes, $w$ in the first term on the right-hand side of Equation (5) represents subgrid vertical velocity derived from TKE and/or OGWs.

16. Line 151: Please specify which scheme is being referenced by "the scheme."

It refers to the LP05 scheme.

17. Lines 150-154: The distinction between the LP05 and K22 schemes is not entirely clear. To enhance clarity, (1) Provide a more explicit description of LP05, similar to the level of detail used for K22. (2) Compare the two schemes in terms of their treatment of vertical velocity, supersaturation parameterization, INP activation efficiency, and aerosol representation.

Thank you very much for your valuable comments. To facilitate better understanding, we have added section 2.2.2 to provide a more explicit description of LP05 equations and section 2.2.3 to compare the two schemes. Overall, both the two schemes use the input of subgrid-scale vertical velocity and aerosols from the host model (e.g., CAM6) and solve the equation of ice supersaturation (Equation 3). However, the INP activation efficiency, and the competition between ice nucleation mechanisms (homogeneous versus heterogeneous) and preexisting ice are treated differently as discussed in section 2.2.3.

18. Lines 150-160: Understanding the differences between LP05 and K22 would be challenging until both schemes, particularly their mathematical formulations, are clearly described.

Thank you very much for your comment. We have introduced a new section 2.2.3 to address the issue in greater detail.

19. Line 167: Does "size" refer to ice crystal diameter? If so, consider specifying.

Thank you very much for your comments. Yes, in this context, the size refers to the diameter of the detected particles. The relevant paragraph has been modified as follows (line 265-267):

"Ice crystals with diameters ranging from 10 to 3000 μm were measured using two-dimensional stereo-imaging probes (2D-S)."

20. Lines 203-205: What is the key distinction between regions 2 and 3? Please clarify.

Thank you very much for your comment. Region 2 is located downwind of the Andes Mountains and Antarctic high plateaus, thus experiencing the additional influence from OGWs on observed cirrus. However, Region 3 is not affected in this way.

21. Line 212: The description of Panel 3-h in Figure 2 is not mentioned in Tables 1 and 2. Should this be added, or did I overlook something?

Thank you very much for your comment. The model results shown in Figure 2h in Figure 2 are based on K22_OGW-Climo and K22_no_TKE-Climo experiments in Table 1. This is noted in Figure 2 caption.

22. Lines 241-248: This paragraph is somewhat unclear. Consider rewording for clarity.

Thank you very much for your comment. We have revised the paragraph as suggested. The relevant paragraph has been modified as follows (line 351-361):

"The K22 scheme simulates higher activated number concentrations of aqueous aerosols for homogeneous nucleation compared to the LP05 scheme. This difference can be attributed to both direct and indirect influences. The direct effect stems from how each scheme represents the competition between nucleated and pre-existing ice crystals. As described in Section 2.2.3, the number of nucleated ice crystals in the LP05 scheme tends to be more suppressed by the competition between pre-existing ice crystals and newly formed ice crystals, compared to the K22 scheme. Consequently, the presence of pre-existing ice crystals leads to fewer ice crystals that are formed, producing overall lower ice number concentrations in the LP05 experiments. The indirect effects are associated with differences in temperatures and vertical velocity fields between the two schemes."

23. Lines 268-269: The conclusion here is difficult to verify based on Section 2.2. Improving Section 2.2 would help clarify why competition between homogeneous

nucleation and pre-existing ice is less pronounced in K22 compared to LP05. Does this primarily depend on supersaturation?

Thank you very much for your comment. Yes, this depends on how the two schemes treat the effect of pre-existing ice on supersaturation. We have included a corresponding section (2.2.3) in the main text to enhance understanding.

24. Lines 275-277: Could you provide supporting evidence for the statement that temperature changes smaller than 0.25°C have a negligible impact on ice number concentration? Do you have references to support this claim? Additionally, is the 0.25°C variation based on monthly-mean data? How does it compare to instantaneous temperature fluctuations?

Thank you very much for your comments. This 0.25°C variation is based on monthly-mean data. The instantaneous temperature fluctuations can be larger. However, the temperature changes are mostly positive in the K22_OGW-Climo experiment compared to the LP05_OGW-Climo experiment. Based on previous studies (Kay & Wood, 2008; Liu & Shi, 2018), 1°C warming could reduce $N_i$ by 5-20%. This temperature increase should suppress the ice nucleation and cannot explain the increased ice number concentration in the K22 scheme. We have revised the sentence to clarify the meaning as follows (line 408-410):

"However, these temperature changes are generally small (typically smaller than $\pm$ 0.25 ℃) and mostly positive, suggesting a suppression of ice nucleation. Therefore, the impact of temperature difference on global $N_i$ is expected to be negative and unlikely to account for a globally significant increase in $N_i$ observed in the K22 scheme (Fig. 2)."

25. Figures S2 and S3: Do LP05 and K22 yield identical latitude distributions for the -40°C layer and tropopause? Why is only one tropopause and -40°C layer shown in these figures? Also, does "corresponding simulation" in the figure captions refer to a specific case?

Thank you very much for your comment. The -40°C layer and the tropopause could be slightly different between the two schemes because of the temperature differences. We have clarified the descriptions of -40°C layer and the tropopause in the figure captions:

"Dashed lines represent the annual mean -40℃ isothermal line, and solid lines are the tropopause in the LP05_OGW-Climo experiment."

26. Lines 282-284: A vertical velocity change of 0.002 m/s seems quite small to significantly influence ice nucleation rates. Could you provide quantitative evidence to support this?

Thank you very much for your comment. Based on previous studies (Hoyle et al., 2005; Kärcher & Lohmann, 2002; Kay & Wood, 2008), an increase of vertical velocity by 0.1 m/s, $N_i$ may increase by a factor of 2-4 depending on temperatures.

27. Lines 289-293: To support the discussion, consider including plots of surface wind speed differences, dust emissions, and deposition rates.

Thank you very much for your comment. Surface wind, dust emission and dust deposition rates are indeed important factors for the distribution of dust number concentrations. Figure R1-9 illustrates the differences in surface wind speed between the K22 and LP05 schemes, showing that the K22 scheme tends to increase surface wind speed over Greenland, Europe, Africa and South America. Figure R1-10 displays the differences in coarse mode dust surface emissions, indicating emissions are enhanced in some regions while suppressed in others. Figures R1-11 and R1-12 present the differences in coarse mode dust wet and dry deposition rates, respectively. The major differences seem that the dry deposition rate of coarse mode dust in the K22 scheme is reduced over dust source regions (e.g., northern Africa, central Asia), which likely leads to the increase in dust number concentrations in the upper troposphere (Figure S4).

[Figure]

Figure R1-9. Differences of annual mean 10 m wind speed (m s$^{-1}$) between K22_OGW-Climo and LP05_OGW-Climo experiments.

[Figure]

Figure R1-10. Differences of coarse mode dust surface emission (kg m$^{-2}$ s$^{-1}$) between K22_OGW-Climo and LP05_OGW-Climo experiments.

[Figure]

Figure R1-11. Differences of coarse mode dust wet deposition rate (kg m$^{-2}$ s$^{-1}$) between K22_OGW-Climo and LP05_OGW-Climo experiments.

[Figure]

Figure R1-12. Differences of coarse mode dust dry deposition (kg m$^{-2}$ s$^{-1}$) between K22_OGW-Climo and LP05_OGW-Climo experiments.

28. Lines 297-299: Please provide plots of heterogeneous and homogeneous nucleation rates to substantiate the discussion.

Thank you very much for your comments. Following your comment, we plot the heterogeneous and homogenous nucleation tendencies (Figure R1-13 and R1-14), similar to the plots in our response to your major comment/question No.4 above. The results show that, in the K22_OGW_Shan-Climo experiment, homogeneous nucleation is enhanced while heterogeneous nucleation is suppressed (Figure R1-13 and R1-14). The related figures have been included in the supplementary materials.

[Figure]

Figure R1-13. Annual ice number tendencies due to homogeneous $\Delta N_{i\_hom}$ and heterogeneous nucleation $\Delta N_{i\_het}$ from 6-year climatology K22_OGW_Shan-Climo experiment at 250 hPa. The second row shows the tendency differences between K22_OGW_Shan-Climo and K22_OGW-Climo experiments.

[Figure]

Figure R1-14. Annual zonal mean ice number tendencies due to homogeneous $\Delta N_{i\_hom}$ and heterogeneous nucleation $\Delta N_{i\_het}$ from 6-year climatology K22_OGW_Shan-Climo experiment. The second row shows the tendency differences between K22_OGW_Shan-Climo and K22_OGW-Climo experiments.

The relevant paragraphs have been modified as follows (line 464-470):

"Further insight into the role of aerosol processes in ice nucleation is provided by the K22_OGW_Shan-Climo experiment, which incorporates an improved treatment of aerosol wet removal by convections based on Shan et al. (2021). In this configuration, dust aerosol concentrations are reduced due to more efficient convective scavenging (Fig. S9), particularly in convectively active low latitude regions. The resulting lower dust number concentrations lead to a reduced heterogeneous nucleation rate, thereby enhancing the homogeneous nucleation rate due to reduced competition from heterogeneous nucleation on dust (Fig. S10 and 11). In this case, improvements in aerosol wet removal may help optimize upper tropospheric aerosol concentrations and can leads to a general increase in $N_i$ (Fig. S12). "

29. Lines 278-300: The logical flow in these paragraphs could be improved. The discussion aims to explain the causes of Ni differences between LP05 and K22, but the explanation is somewhat difficult to follow. In particular, the role of activated INP fraction (Φ) in Lines 296-287 is unclear. Since Φ is influenced by vertical velocity, temperature, and water vapor, could you clarify why it is considered an independent factor driving Ni? Consider: (1) Listing all key factors influencing Ni concentration. (2) Comparing these factors between LP05 and K22.

Thank you very much for your comments. The relevant paragraphs have been modified to list all key factors influencing $N_i$ concentrations and then compare these factors between the two ice nucleation schemes as follows (line 401-436):

"To analyze the factors driving differences in $N_i$ between the LP05 and K22 schemes, several key variables should be considered. These factors include temperature, which affects ice nucleation thresholds and saturation vapor pressure; subgrid-scale vertical velocity, which determines the supersaturation necessary for ice formation; and dust aerosol number concentration, along with the fraction of activated INPs (Φ), which together determine the number of heterogeneously nucleated ice crystals.

In high latitudes, temperature increases in the upper troposphere are found in the K22_OGW-Climo experiment compared to the LP05_OGW-Climo experiment (Fig. S2), likely due to localized warming associated with increased cirrus cloud occurrence (Fig. S1). However, these temperature changes are generally small (typically smaller than ± 0.25 ℃) and mostly positive, suggesting a suppression of ice nucleation. Therefore, the impact of temperature difference on global $N_i$ is expected to be negative and unlikely to account for a globally significant increase in $N_i$ observed in the K22 scheme (Fig. 2).

Similarly, subgrid-scale vertical velocity increases in the K22_OGW-Climo experiment compared to the LP05_OGW-Climo experiment, particularly in the upper troposphere at mid- and high latitudes (Fig.S3). While these changes may enhance ice nucleation locally, their overall impact on $N_i$ remains limited, as vertical velocity changes are generally small (less than ±0.002 m s$^{-1}$) in most regions. Therefore, they are unlikely to explain the globally significant increase in $N_i$ simulated in the K22 scheme (Fig. 2).

The most substantial differences in $N_i$ between the two schemes arise from microphysical processes, particularly those governing heterogeneous ice nucleation. Both the K22 and LP05 schemes account for the activation of coarse mode dust particles, but

the K22 scheme simulates higher dust aerosol number concentrations, especially in the upper troposphere (Fig. S4). This enhancement is likely driven by changes in large scale circulation patterns and surface wind fields resulting from differences in the applied ice nucleation schemes, which influence both dust emission and atmospheric transport pathway. As a result, the K22 scheme shows an increase in ice number concentration nucleated from dust particles heterogeneously, as shown in Fig. 3c and 3d. The activated INP fraction Φ also plays a crucial role in controlling heterogeneous nucleation. While Φ depends on local thermodynamic conditions, such as temperature, vertical velocity, and supersaturation in the LP05 scheme, the K22 scheme simplifies this dependence, with Φ relying on supersaturation only. Differences in the treatment of Φ, combined with elevated dust concentrations in the K22 scheme may influence heterogeneous nucleation on coarse mode dust. However, since the number of coarse mode dust is limited (~10-30 $L^{-1}$) in the upper troposphere (Fig. S4), even if all the dust particles are nucleated heterogeneously to form ice crystals, their contribution to increased $N_i$ will not reach the levels (~100 $L^{-1}$) observed in the K22 scheme. Therefore, these two factors are unlikely to explain the globally significant increase in $N_i$ seen in the K22 scheme compared to the LP05 scheme (Fig. 2a and Fig. 2b). This also implies that competition between preexisting ice and new ice nucleation is a more dominant factor influencing the simulated $N_i$. ”

30. Line 326: Does OGW always increase Ni? Consider revising for clarity.

Thank you very much for your comment. OGWs influence the vertical velocity, which determines ice supersaturation. In the SPARTICUS orographic cirrus, $N$i is dominantly formed from homogeneous nucleation, OGWs can increase vertical velocity and potentially increase $N$i. However, in cirrus cases where $N$i can be influenced by many other processes, OGWs do not always lead to an increase in $N$i.

The relevant paragraph has been modified as follows (line 537-540):

“Fig. 5b shows that in both LP05 and K22 schemes, the changes in $N_i$ ($\Delta N_i$) due to OGWs are always positive and larger than those from the other two sources in these cirrus clouds. This indicates that OGWs play a significant role in enhancing the formation of ice crystals in cirrus clouds identified as orographic cirrus during the observed five-days period. ”

31. Lines 332-339: Additional evidence is needed to support this statement. Could you provide plots showing the changes in homogeneous and heterogeneous freezing rates to illustrate their respective contributions? Also, please consider refining the wording, as the term “contribution” can imply either a positive or negative effect. I assume “contribution and inhibition” refer to invigoration and suppression, respectively. Could you confirm?

Thank you very much for your valuable comments. We use the $\Delta N_i$ (similar to ice number tendency) to evaluate the contribution, and we do not show the changes from

homogeneous and heterogeneous freezing rates, because we are also interested in the effect of convective detrainment and separate effects of OGWs and turbulence on $N_i$ (through ice nucleation). We have improved the wording and used "enhancement" instead of "contribution" when the effects are positive.

The relevant paragraph has been modified as follows (line 537-551):

"Fig. 5b shows that in both LP05 and K22 schemes, the changes in $N_i$ ($\Delta N_i$) due to OGWs are always positive and larger than those from the other two sources in these cirrus clouds. This indicates that OGWs play a significant role in enhancing the formation of ice crystals in cirrus clouds identified as orographic cirrus during the observed five-days period. Particularly in regions with temperatures below 215 K, where both schemes simulate their highest $N_i$ peaks, $\Delta N_i$ due to OGWs peaks positively at the corresponding temperatures. This suggests that OGW-induced ice crystals enhance the overall $N_i$ in these cirrus clouds. Detrained and turbulence-induced $\Delta N_i$ values show different signs, fluctuating between positive and negative at different temperatures, indicating that the effects of the other two sources are uncertain and vary between the two schemes. In the LP05 scheme, detrained and turbulence-induced $\Delta N_i$ values are generally negative, suggesting that ice crystals from both detrainment and turbulence tend to inhibit $N_i$. In contrast, the K22 scheme exhibits varied detrained and turbulence-induced $\Delta N_i$ values, with stronger fluctuations between positive and negative with temperature, indicating that these sources can either enhance or inhibit $N_i$. Notably, the positive $\Delta N_i$ values in detrained and turbulence-induced ice crystals are smaller in the LP05 scheme, suggesting stronger competition (inhibition effects) between ice sources in the LP05 scheme. "

**32. Lines 340-341: Could you clarify the definition of Dnum?**

Thank you very much for your comments. Dnum is defined as the number weighted diameter. Assuming the ice size distribution follow gamma distribution with coefficients $N_0$ and $\lambda$, ($D_{num} = \frac{\int_0^\infty N_0 D e^{-\lambda D} dD}{\int_0^\infty N_0 e^{-\lambda D} dD}$).

To better address the issue, the relevant paragraph has been modified as follows (line 558-559):

"Regarding the simulated number weighted diameter of ice crystals $D_{num}$ in the LP05 and K22 experiments (Fig. S15 and S16),…. "

**33. Line 354: Dnum should be defined upon first mention for clarity.**

Thank you very much for your comments. The $D_{num}$ is first defined in Section 3.1 at line 257. The relevant sentence is as follows (line 283-286) :

"Additionally, the microphysical properties (such as ice number $N_i$, ice water content IWC and number-weighted diameters $D_{num}$) of ice crystals with diameters larger than 20 μm from CAM6 results are derived using the size cut method described by Eidhammer et

al. (2014), consistent with the measurements obtained by the 2D-Stereo Particle Probe (2D-S) but excluding the first size bin. "

34. Lines 356-357: "This characteristic helps explain why the K22 scheme results in increased cloud frequency compared to the LP05 scheme." Could you provide a more detailed explanation? Specifically, why do cirrus clouds with a higher number of smaller ice crystals lead to increased cloud frequency?

Thank you very much for your comment. We have clarified the issue as follows (line 815-817):

"This can be due to the presence of smaller ice crystals in the K22 scheme, which have smaller fall speeds, allowing them to travel over broader regions before completely sublimated. "

35. Lines 376-377: Please specify the number of simulation and observational samples used in the analysis. It would also be helpful to include the corresponding sample sizes for SPARTICUS and the experiments.

Thank you very much for your comments. We have included additional information about sample sizes in the article. There are 53987 (6236) data samples in the SPARTICUS observational and simulation datasets (in five days identified as orographic cirrus events). The datasets during the ORCAS campaign include 341410 samples. The relevant paragraphs have been modified as follows (line 268-270):

"A total of 6236 data points are available in both observational and simulated datasets during the five days identified as orographic cirrus events (Muhlbauer et al., 2014)."

 (line 606-608):
"In Region 1,... The dataset used in the analysis includes 83559 data points."

 (line 640-642):
"Region 2, ... The dataset used in the analysis includes 146139 data points."

 (line 689-691):
"In Region 3, ... There are 111712 data points used in the analysis."

36. Lines 386-387: "However, in the LP05 scheme, ice crystals due to OGWs and detrainment tend to inhibit the formation of simulated Ni, whereas their effects are minimal in the K22 scheme." To support this statement, I recommend including plots of homogeneous and heterogeneous freezing rates. Similar evidence is needed for the statements in Lines 387-391. Additionally, aerosol number concentrations (for both sulfate and dust) should be provided to help explain the differences in Ni.

Thank you very much for your comments. Fig. 7b shows the $\Delta N_i$ (i.e., ice number tendencies) due to ice nucleation from OGWs and turbulence and due to detrainment. We would like to show the competition between different ice sources, i.e., ice nucleation by OGWs and turbulence and by convective detrainment. We believe that this is a clearer explanation than showing the ice nucleation rates from homogeneous and heterogeneous freezing.

The relevant paragraph has been modified as follows (line 629-634):

"At the 210 K level, the overwhelmingly positive $\Delta N_i$ values due to turbulence in both schemes suggests that turbulence-induce ice crystals are the primary contributors (Fig. 7b). However, in the LP05 scheme, $\Delta N_i$ values due to OGWs are negative, suggesting that OGW-induced ice crystals tend to inhibit ice crystal formation. In contrast, their impacts (OGW-induced $\Delta N_i$) are minimal (~0) in the K22 scheme, indicating no evident inhibitory effect. In addition, both schemes simulate generally negative $\Delta N_i$ values due to detrainment, implying that detrained ice crystals tend to suppress further ice formation. "

 (line 543-551):

    "Detrained and turbulence-induce $\Delta N_i$ values show different signs, fluctuating between positive and negative, indicating that the effects of the other two sources are uncertain and vary between the two schemes. In the LP05 scheme, generally negative detrained and turbulence-induce $\Delta N_i$ values suggest that ice crystals from both detrainment and turbulence tend to inhibit $N_i$. In contrast, the K22 scheme exhibits varied detrained and turbulence-induce $\Delta N_i$ values, with stronger fluctuations between positive and negative, indicating that these sources can either enhance or inhibit $N_i$. .Notably, the number of positive $\Delta N_i$ values in detrained and turbulence-induced ice crystals is smaller in the LP05 scheme, suggesting stronger competitive (inhibition effects) between ice sources in the LP05 scheme. "

 (line 652-664):

    "In Fig. 8a, similar to Region 1, multiple high $N_i$ peaks again correspond to different primary $\Delta N_i$ contributors, suggesting a multilayer structure of cirrus clouds in Region 2. Near 215 K, the OGW experiments in both schemes simulate high $N_i$ peaks that closely match the observed high peak near 218 K. The corresponding positive OGW-induced $\Delta N_i$ values in both schemes (Fig. 8b) suggest that a large portion of these ice crystals are generated by OGWs originating from mountains and high plateaus. The contributions from other sources (detrained $\Delta N_i$ and turbulence-induced $\Delta N_i$) differ between the schemes. In the LP05 scheme, generally positive detrained $\Delta N_i$ and fluctuating turbulence-induced $\Delta N_i$ near 215K suggest an enhancing role from detrained ice crystals and a mix of enhancing and inhibiting effects from turbulence-induced ice crystals. In contrast, the K22 scheme exhibits negative $\Delta N_i$ values for both sources, indicating overall inhibition effects. These findings imply that the $N_i$ peaks around 215 K are strongly related to the mountainous terrain upwind of Region 2. "

 (line 674-688):

"In the lower part of cirrus clouds (T > 225K), negative $\Delta N_i$ values of all three ice crystal sources in the LP05 scheme suggest universal competition. In contrast, in the K22 scheme, only detrained $\Delta N_i$ values are negative, implying inhibition effects, while positive $\Delta N_i$ values from OGWs and turbulence suggest these ice crystals enhance $N_i$. . The fact that no single $\Delta N_i$ value is positive in both schemes may suggest that the dominant ice source is missing from the model. Previous studies have highlighted the importance of additional ice nucleation mechanisms, such as frontal gravity waves, in cirrus formation over oceans, and identified crucial INPs including dust, metallic particles, soot and biological materials (Fan et al., 2016; Froyd et al., 2022; Heymsfield et al., 2017; Kärcher & Ström, 2003; Knopf & Alpert, 2023). However, in CAM6, only orographic gravity waves are included in ice nucleation scheme, and only coarse mode dust is considered as INPs. In addition, other important $N_i$ source and sink processes, such as secondary ice production, ice sublimation and sedimentation should be examined. Future studies are therefore necessary to incorporate these potential dynamic and microphysical sources to improve simulations of cirrus clouds over oceanic regions. "

37. Line 403: The phrase "predominantly generated from OGWs" may not be entirely appropriate, as turbulence-induced increases in Ni in the K22 scheme are also significant. Please consider rewording this statement to more accurately reflect the relative contributions of different mechanisms.

Thank you very much for your comments. We have modified the relevant sentence as follows (line 656-658):

"The corresponding positive OGW-induced $\Delta N_i$ values in both schemes (Fig. 8b) suggest that a large portion of these ice crystals are generated by OGWs originating from mountains and high plateaus. "

38. Lines 411-412: The phrase "wider spread of ice crystals" is somewhat unclear. Are you referring to ice redistribution due to advection? Please clarify.

Thank you very much for your comments. We have modified the relevant sentences as follows (line 670-673):

"In the K22 scheme, however, the high $N_i$ (>100 $L^{-1}$) extends over a larger area, facilitating interaction and competition between OGW-induced ice sources with other ice sources even far from the mountainous regions."

39. Line 430: Could you clarify why this does not qualify as a clean oceanic environment?

Thank you very much for your comment. This is primarily an oceanic environment, and in our analysis, we do not classify it as a "clean" or "polluted" environment because we do not have relevant aerosol or other tracer gas observational data to validate.

40. Lines 430-431: "At the cloud top, ice crystals due to turbulence make the most significant contributions to the simulated Ni peaks when T < 210 K in both schemes (Fig. 9b)." This sentence is unclear. How was Ni at the cloud top identified?

Thank you very much for your comments. We have modified the relevant sentence as follows (line 709-711):

"At low temperature levels (T < 209 K), both schemes exhibit positive turbulence-induced $\Delta N_i$ values, suggesting that ice crystals due to turbulence make the most contributions to the $N_i$ at these cold temperatures (Fig. 9b). "

41. Lines 431-435: A general comment: The analysis of the "main ice source" may be affected by uncertainties in three drivers: ice crystal detrainment from deep convection, TKE, and OGWs. If this is the case, it would be beneficial to discuss the uncertainties in these drivers in the study, perhaps in the discussion section.

Thank you very much for your comments. We completely agree that the "main ice source" may be affected by the uncertainties in the three drivers: ice crystal detrainment from deep convection, TKE, and OGWs. We have added some sentences in discussions section 5 (line 880-884):

"Further studies should also consider incorporating additional dynamic processes, such as frontal and convective gravity waves (Yook et al., 2025). In addition to gravity waves, uncertainties in the representation of other drivers of ice sources, such as turbulence and convective detrainment, should be reduced. Recent incorporations of convective cloud microphysics in deep convection (Lin et al., 2021; Song & Zhang, 2011) should help to reduce the uncertainty in detrained ice properties. "

42. Lines 436-449: The discussion in this section could be better structured. First, the uncertainties in the INP activation rate arise from two primary factors: (1) supersaturation and (2) aerosol concentrations. The detrainment, OGW, and TKE influence supersaturation, which may only partially explain the biases in Ni. Second, other source and sink terms beyond INP activation, such as secondary ice production, ice sublimation, and sedimentation, may also play a significant role. Could you clarify how these processes contribute?

Thank you very much for your comments. We agree with the reviewer regarding the processes influencing ice number $N_i$ in cirrus clouds, first, ice formation depending on supersaturation and aerosols, and then ice evolution depending on secondary ice production, ice sublimation and sedimentation. We have modified the relevant paragraphs to better structure the discussion as follows (line 712-727):

"In the lower levels of cirrus (T > 227 K), most of the simulated $N_i$ peaks occur (Fig. 9a). At these temperatures, turbulence-induced $\Delta N_i$ values are mostly positive and generally exceed OGW-induced and detrained $\Delta N_i$ values in both schemes, suggesting a strong enhancement of $N_i$ from turbulence. However, OGW-induced and detrained $\Delta N_i$ values differ between the two schemes. In the K22 scheme, positive OGW-induced and detrained $\Delta N_i$ values suggest significant enhancements to $N_i$ from OGWs and detrainment. In contrast, the LP05 scheme shows large variability, with OGW-induced and detrained $\Delta N_i$ values fluctuating between positive and negative, indicating more complex and varied effects from these ice sources in the simulations.

Numerous studies have demonstrated that turbulence from CLUBB-TKE can hardly predict perturbations from gravity waves (Golaz et al., 2002a, 2002b; Huang et al., 2020). To accurately simulate cirrus clouds over oceans in Region 3, it is necessary to incorporate representations of other key dynamic drivers for ice nucleation, such as frontal and convective gravity waves. It is also important to incorporate key INPs (e.g., marine organic aerosols) besides mineral dust into ice nucleation schemes. Other source and sink terms beyond ice nucleation, such as secondary ice production, ice sublimation, and sedimentation, may also play a significant role in influencing the $N_i$ evolution over oceans. "

43. Line 451: "Both K22 and LP05 schemes can effectively simulate the dominant ice sources." Could you confirm whether the dominant ice source in this study is INP activation? As far as I understand, secondary ice production may surpass INP activation in driving Ni.

Thank you very much for your comment. We agree that secondary ice production is a very important ice source. We have modified the sentence to avoid any misunderstandings as follows (line 740-741):

"Both K22 and LP05 schemes can effectively simulate the ice nucleation as a dominant ice source in orographic cirrus clouds, though they exhibit different effects from other ice sources on simulated $N_i$. "

44. Lines 458-462: While OGWs are known to induce high supersaturation conducive to ice formation, the large increase in Ni may result from both INP activation and secondary ice production (e.g., ice multiplication during solution droplet freezing). Is there any evidence indicating that INP activation is the dominant process in this case?

Thank you very much for your comment. It is known that secondary ice production (SIP) can lead to an increase in ice number by several orders of magnitudes over $N_i$ from primary ice nucleation. Our simulations without considering SIP in the model can reproduce the $N_i$ in orographic cirrus observed in SPARTICUS reasonably well. Thus, we don't expect that the SIP is the main factor for the observed $N_i$ here. Furthermore, ice shattering during solution droplet freezing requires drizzle size drops (>100 $\mu$m) at much larger temperatures (Luke et al., 2021), which don't exist in these cold cirrus clouds.

45. Lines 469-470: "OGW-induced ice crystals are the dominant contributors in these 16 days of cirrus clouds (Fig. 10b)." Based on Figure 10b, it appears that OGWs primarily dominate Ni in the K22 scheme but not in LP05. Additionally, in the K22 scheme, detrainment appears to contribute comparably to OGWs. Would you consider revising this statement to better reflect these findings?

Thank you very much for your comment. We found an error in our plotting script for Figure 10b. After correcting it, detrainment plays a much smaller role compared to OGWs in the K22 scheme. OGWs still dominate $N_i$ in LP05 compared to other contributors (detrainment and turbulence), although the magnitude from OGWs is smaller in LP05 than in K22.

46. Lines 473-474: More evidence is needed to support the assertion that the assumed detrainment ice size is inappropriate. Ice crystals above the -40°C layer are typically smaller than 50 µm, correct? If so, please provide references or observational data to substantiate this claim. Additionally, since both the K22 and LP05 schemes employ the same ice size assumption, why does LP05 underestimate Ni? It seems that Lines 340-344 attempt to explain this, but the connection between ice nucleation competition and Dnum changes is unclear. Could you provide a clearer explanation, particularly regarding the underlying physical mechanisms?

Thank you very much for your comments. We agree that there are uncertainties on assumed detrained ice size. Unfortunately, we find limited references in literature that directly address detrained ice sizes. It is true that both the K22 and LP05 schemes use the same assumed ice size of 50 µm for detrained ice. Because detrainment plays a minor role in the orographic cirrus identified in SPARTICUS (see corrected Figure 10b), we have deleted these sentences to avoid confusion.

47. Lines 497-498: Further evidence is needed to support the claim that smaller ice crystals have longer lifetimes. While smaller ice crystals indeed fall more slowly, their larger collective surface area may enhance sublimation in subsaturated conditions. Could you compare these competing mechanisms?

Thank you very much for your comment. A small ice crystal falls more slowly than a large one and typically has smaller surface area for sublimation. These two factors allow small ice crystals to remain in the atmosphere for longer periods. However, if ice water content of ice crystals is the same, the collective surface area of fewer large ice crystals would be smaller than that of small ones. The total sublimation rate of the large ones would be therefore lower than that of small ones. However, the role of sedimentation appears to be more important, because if ice crystals fall slowly they tend to stay within clouds and thus less subject to the sublimation in subsaturated conditions.

We have modified the sentence in the revision (line 815-817):

"This can be due to the presence of smaller ice crystals in the K22 scheme, which have lower fall speeds, allowing them to travel over broader regions before completely sublimated. "

48. Lines 498-500: The changes in cloud frequency and circulation dynamics do not appear to be particularly striking. Since the vertical velocity used in the INP scheme corresponds to subgrid-scale processes, while the ascending motion of large-scale circulation is represented by grid-scale vertical velocity, how do you reconcile this difference in scale?

Thank you very much for your comments. The magnitude of large-scale circulation is typically on the order of 0.001 m/s, while subgrid-scale vertical velocities generally range from 0.01 to 0.1 m/s. In this paper, we emphasize that the indirect influence of $N_i$ between different nucleation schemes are not driven by changes in subgrid-scale vertical velocity. We modified the sentence to make it clearer (line 817-820):

"An increase in cloud frequency may induce changes in global temperature, potentially affecting subgrid-scale vertical velocity, thereby impacting ice nucleation. However, these factors are not the key factors that cause the significant increase in $N_i$. "

49. Lines 501-503: As mentioned here, there are clear differences in dust concentrations between the K22 and LP05 schemes. Could you also discuss the dust differences in the nudging runs? In Sections 4.2-4.3, the differences in Ni between K22 and LP05 are attributed primarily to detrainment, OGWs, and TKE. However, the impact of dust concentration differences on Ni is not discussed. Could you clarify why dust differences were not considered in these sections?

Thank you very much for your comment. As shown in Fig. R1-1 above, the nudged experiments during the SPARTICUS campaign indicate that coarse mode dust number concentration in the K22 scheme tends to be higher than that in the LP05 scheme. However, dust number concentrations appear to be very low ($< 1 \ \text{L}^{-1}$) and thus

heterogeneous nucleation on dust tends to be less important for $N_i$ compared to other factors.

50. Lines 507-508: See my earlier comment on Line 45. It does not appear that OGWs dominate Ni in both the LP05 and K22 schemes. Additionally, conclusions drawn from short simulations covering only a few days may not be sufficiently robust. Could you comment on the limitations of these short-term results?

Thank you very much for your comment. We have modified the relevant sentences as follows (line 827-829):

"Both the LP05 and K22 schemes identify OGWs as the dominant ice crystal source in orographic cirrus clouds observed during SPARTICUS, but the LP05 scheme exhibits greater competition from detrainment and turbulence sources than the K22 scheme. "

51. Lines 521-523: While it is clear that low-level moisture is generally higher over the ocean than over land, I am particularly interested in how significant this difference is in the upper troposphere and lower stratosphere. Could you provide a moisture profile plot to illustrate this difference?

Thank you very much for your comments. We have included specific humidity (Q) profiles over land and oceans from the OGW experiments in the two schemes during the ORCAS and SPARTICUS campaigns. The Q vertical profiles are similar in both schemes (Fig. R1-16 and R1-17). During the ORCAS campaign, the Q profiles over land and ocean are comparable, with slightly higher Q over land near 600hPa, indicating that high-level moisture over land is not necessarily lower than over ocean.

[Figure]

Fig. R1-16. Average specific humidity (Q) profiles over land and ocean in LP05_OGW-OR and K22_OGW-OR experiments along the flight tracks during the ORCAS campaign.

[Figure]

Fig. R1-17. Average specific humidity (Q) profiles over Land in LP05_OGW-SP and K22_OGW-SP experiments along the flight tracks during the SPARTICUS campaign.

Reference:

Beall, C. M., Hill, T. C. J., DeMott, P. J., Köneman, T., Pikridas, M., Drewnick, F., Harder, H., Pöhlker, C., Lelieveld, J., Weber, B., Iakovides, M., Prokeš, R., Sciare, J., Andreae, M. O., Stokes, M. D., & Prather, K. A. (2022). Ice-nucleating particles near two major dust source regions. *Atmos. Chem.Phys.*, *22*(18), 12607-12627. https://doi.org/10.5194/acp-22-12607-2022

Chen, J., Wu, Z., Gong, X., Qiu, Y., Chen, S., Zeng, L., & Hu, M. (2024). Anthropogenic Dust as a Significant Source of Ice-Nucleating Particles in the Urban Environment. *Earth's Future*, *12*(1), e2023EF003738. https://doi.org/https://doi.org/10.1029/2023EF003738

The Earth's Energy Budget, Climate Feedbacks and Climate Sensitivity. (2023). In C. Intergovernmental Panel on Climate (Ed.), *Climate Change 2021 – The Physical Science Basis: Working Group I Contribution to the Sixth Assessment Report of the Intergovernmental Panel on Climate Change* (pp. 923-1054). Cambridge University Press. https://doi.org/DOI: 10.1017/9781009157896.009

Eidhammer, T., Morrison, H., Bansemer, A., Gettelman, A., & Heymsfield, A. (2014). Comparison of ice cloud properties simulated by the Community Atmosphere Model (CAM5) with in-situ observations. *Atmospheric Chemistry & Physics*, *14*(18). https://doi.org/https://doi.org/10.5194/acp-14-10103-2014

Fan, J., Wang, Y., Rosenfeld, D., & Liu, X. (2016). Review of Aerosol-Cloud Interactions: Mechanisms, Significance and Challenges. *Journal of the Atmospheric Sciences*, *73*. https://doi.org/10.1175/JAS-D-16-0037.1

Froyd, K. D., Yu, P., Schill, G. P., Brock, C. A., Kupc, A., Williamson, C. J., Jensen, E. J., Ray, E., Rosenlof, K. H., Bian, H., Darmenov, A. S., Colarco, P. R., Diskin, G. S., Bui, T., & Murphy, D. M. (2022). Dominant role of mineral dust in cirrus cloud formation revealed by global-scale measurements. *Nature Geoscience*, *15*(3), 177-183. https://doi.org/10.1038/s41561-022-00901-w

Gettelman, A., Liu, X., Ghan, S. J., Morrison, H., Park, S., Conley, A. J., Klein, S. A., Boyle, J., Mitchell, D. L., & Li, J.-L. F. (2010). Global simulations of ice nucleation and ice supersaturation with an improved cloud scheme in the Community Atmosphere Model. *Journal of Geophysical Research: Atmospheres*, *115*(D18). https://doi.org/https://doi.org/10.1029/2009JD013797

Golaz, J.-C., Larson, V. E., & Cotton, W. R. (2002a). A PDF-Based Model for Boundary Layer Clouds. Part I: Method and Model Description. *Journal of the Atmospheric Sciences*, *59*(24), 3540-3551 , ISSN = 0022-4928 , DOI = https //doi.org/3510.1175/1520-0469. https://journals.ametsoc.org/view/journals/atsc/59/24/1520-0469_2002_059_3540_apbmfb_2.0.co_2.xml

Golaz, J.-C., Larson, V. E., & Cotton, W. R. (2002b). A PDF-Based Model for Boundary Layer Clouds. Part II: Model Results. *Journal of the Atmospheric Sciences*, *59*(24), 3552-3571 , ISSN = 0022-4928 , DOI = https //doi.org/3510.1175/1520-0469. https://journals.ametsoc.org/view/journals/atsc/59/24/1520-0469_2002_059_3552_apbmfb_2.0.co_2.xml

Heymsfield, A. J., Krämer, M., Luebke, A., Brown, P., Cziczo, D. J., Franklin, C., Lawson, P., Lohmann, U., McFarquhar, G., Ulanowski, Z., & Van Tricht, K. (2017). Cirrus Clouds. *Meteorological Monographs*, *58*, 2.1-2.26. https://doi.org/https://doi.org/10.1175/AMSMONOGRAPHS-D-16-0010.1

Hoyle, C. R., Luo, B. P., & Peter, T. (2005). The Origin of High Ice Crystal Number Densities in Cirrus Clouds. *Journal of the Atmospheric Sciences*, *62*(7), 2568-2579 , ISSN = 0022-4928 , DOI = https //doi.org/2510.1175/jas3487.2561. https://journals.ametsoc.org/view/journals/atsc/62/7/jas3487.1.xml

Huang, M., Xiao, H., Wang, M., & Fast, J. D. (2020). Assessing CLUBB PDF Closure Assumptions for a Continental Shallow-to-Deep Convective Transition Case Over Multiple Spatial Scales. *Journal of Advances in Modeling Earth Systems*, *12*(10), e2020MS002145. https://doi.org/https://doi.org/10.1029/2020MS002145

Kärcher, B., & Lohmann, U. (2002). A parameterization of cirrus cloud formation: Homogeneous freezing of supercooled aerosols. *Journal of Geophysical Research: Atmospheres*, *107*(D2), AAC 4-1-AAC 4-10. https://agupubs.onlinelibrary.wiley.com/doi/abs/10.1029/2001JD000470

Kärcher, B., & Ström, J. (2003). The roles of dynamical variability and aerosols in cirrus cloud formation. *Atmos. Chem. Phys.*, *3*(3), 823-838. https://acp.copernicus.org/articles/3/823/2003/

Kay, J. E., & Wood, R. (2008). Timescale analysis of aerosol sensitivity during homogeneous freezing and implications for upper tropospheric water vapor

budgets. *Geophysical Research Letters*, *35*(10).
https://doi.org/https://doi.org/10.1029/2007GL032628

Knopf, D. A., & Alpert, P. A. (2023). Atmospheric ice nucleation. *Nature Reviews Physics*, *5*(4), 203-217. https://doi.org/10.1038/s42254-023-00570-7

Koop, T., Luo, B., Tsias, A., & Peter, T. (2000). Water activity as the determinant for homogeneous ice nucleation in aqueous solutions. *Nature*, *406*(6796), 611-614. https://doi.org/10.1038/35020537

Lin, L., Fu, Q., Liu, X., Shan, Y., Giangrande, S. E., Elsaesser, G. S., Yang, K., & Wang, D. (2021). Improved Convective Ice Microphysics Parameterization in the NCAR CAM Model. *Journal of Geophysical Research: Atmospheres*, *126*(9), e2020JD034157. https://doi.org/https://doi.org/10.1029/2020JD034157

Liu, X., & Penner, J. (2005). Ice nucleation parameterization for global models. *Meteorologische Zeitschrift*, *14*, 499-514 , DOI = https //doi.org/410.1127/0941-2948/2005/0059.

Liu, X., & Shi, X. (2018). Sensitivity of Homogeneous Ice Nucleation to Aerosol Perturbations and Its Implications for Aerosol Indirect Effects Through Cirrus Clouds. *Geophysical Research Letters*, *45*(3), 1684-1691. https://doi.org/https://doi.org/10.1002/2017GL076721

Lyu, K., Liu, X., Bacmeister, J., Zhao, X., Lin, L., Shi, Y., & Sourdeval, O. (2023). Orographic Cirrus and Its Radiative Forcing in NCAR CAM6. *Journal of Geophysical Research: Atmospheres*, *128*(10), e2022JD038164. https://agupubs.onlinelibrary.wiley.com/doi/abs/10.1029/2022JD038164

Muhlbauer, A., Ackerman, T. P., Comstock, J. M., Diskin, G. S., Evans, S. M., Lawson, R. P., & Marchand, R. T. (2014). Impact of large-scale dynamics on the microphysical properties of midlatitude cirrus. *Journal of Geophysical Research: Atmospheres*, *119*(7), 3976-3996. https://agupubs.onlinelibrary.wiley.com/doi/abs/10.1002/2013JD020035

Shan, Y., Liu, X., Lin, L., Ke, Z., & Lu, Z. (2021). An Improved Representation of Aerosol Wet Removal by Deep Convection and Impacts on Simulated Aerosol Vertical Profiles. *Journal of Geophysical Research: Atmospheres*, *126*(13), e2020JD034173. https://agupubs.onlinelibrary.wiley.com/doi/abs/10.1029/2020JD034173

Shi, X., Liu, X., & Zhang, K. (2015). Effects of pre-existing ice crystals on cirrus clouds and comparison between different ice nucleation parameterizations with the Community Atmosphere Model (CAM5). *Atmos. Chem. Phys.*, *15*(3), 1503-1520. https://acp.copernicus.org/articles/15/1503/2015/

Song, X., & Zhang, G. J. (2011). Microphysics parameterization for convective clouds in a global climate model: Description and single-column model tests. *Journal of Geophysical Research: Atmospheres*, *116*(D2). https://doi.org/https://doi.org/10.1029/2010JD014833

Tobo, Y., Adachi, K., DeMott, P., Hill, T., Hamilton, D., Mahowald, N., Nagatsuka, N., Ohata, S., Uetake, J., Kondo, Y., & Koike, M. (2019). Glacially sourced dust as a potentially significant source of ice nucleating particles. *Nature Geoscience*, *12*, 1-6. https://doi.org/10.1038/s41561-019-0314-x

Yook, S., Solomon, S., Weimer, M., Kinnison, D., Garcia, R., & Stone, K. (2025). Implementation of Sub-Grid Scale Temperature Perturbations Induced by Non-

Orographic Gravity Waves in WACCM6. *Journal of Advances in Modeling Earth Systems*, *17*. https://doi.org/10.1029/2024MS004625

---

## Author Comment (AC2)

We thank the two anonymous reviewers for their constructive comments. Below, we explain how the comments are addressed and make notes of the revisions in the revised manuscript. The reviewers' comments are in blue color. Our replies are in black, and our corresponding revisions in the manuscript are in red (line numbers are based on the tracked version of the revised manuscript).

Review of "Exploring Sources of Ice Crystals in Cirrus Clouds: Comparative Analysis of Two Ice Nucleation Schemes in CAM6" by Lyu et al. [Research Article, egusphere-2024-4144]

This study coupled a novel ice nucleation parameterization scheme based on Karcher (2022) into CAM6 and compared its representation of cirrus ice cloud microphysics against the default Liu and Penner (2005) scheme, with a particular focus on ice sources in cirrus cloud formation. The authors conducted a thorough assessment using both long-term simulations and case studies from the SPARTICUS and ORCAS campaigns. Their findings revealed several similarities between the two schemes, such as the climatological location of orographic gravity wave (OGW)-induced ice crystals and the same dominant source (OGW-induced) for orographic cirrus. Notable differences were also identified, primarily attributed to the distinct nucleation/competition mechanisms within the two schemes. Overall, the manuscript is well written, but its structure could be improved for better readability. For example, the excessive use of short paragraphs disrupts the flow, and combining some of them could enhance clarity. This work holds significant potential for advancing ice cloud simulations, particularly in refining parameter tuning and improving the representation of competition mechanisms. However, my major concern is the lack of sufficient physical explanations and robust evidence for the model biases and the differences found between the two schemes. If these issues can be addressed, I believe this paper will be well-suited for publication in ACP.

Thank you very much for your helpful and constructive comments. Following your comments, we have provided more physical explanations and robust evidence for the model biases and the differences found between the two schemes. We also improved our writing by combining some of the short paragraphs which are relevant.

Major comments:

1. A key concept in this study is the competition between homogeneous and heterogeneous freezing in ice cloud formation. The authors argued that the competition is stronger in the LP05 scheme than in K22 due to differences in their parameterization of homogeneous nucleation occurrence. However, this claim appears to be more of an assumption than a rigorously validated conclusion, as it is not directly substantiated from the parameterization formulas (not shown by the authors). The authors have used this assumption multiple times (e.g., Lines 246-248 and 268-270) to explain discrepancies in simulated ice cloud microphysics between the two schemes. I think a more appropriate way would be to first make this assumption explicit and then examine it using supporting evidence from simulation results.

The authors found that fewer new ice crystals form in LP05 with the presence of pre-existing ice crystals (Line 247), which aligns with the assumption of stronger competition in LP05. However, a critical underlying assumption is that both schemes should have a similar or comparable number concentration of pre-existing ice crystals. If the LP05 experiments contain a higher concentration of pre-existing ice crystals than K22, it becomes difficult to determine whether the reduction in new ice formation is genuinely due to stronger competition in LP05 or a result of differing initial conditions. To address this issue, the authors should ensure that the number concentration of pre existing ice crystals is close across experiments or, at the very least, discuss the potential influence of variations in pre-existing ice concentrations on their results.

Additionally, the proposed indirect explanation for the increase in ice number concentration in K22 is not sufficiently substantiated. For example, the authors did not show evidence on how the changed circulation dynamics impact the sub-grid turbulence, making this explanation remain speculative rather than a well-supported conclusion.

Thank you very much for your comments.

First, we appreciate the reviewer's comment regarding that the competition among ice sources is stronger in the LP05 scheme than in the K22 scheme. While this assessment may appear to rely on assumptions, it is actually supported by the relevant figures, specifically Fig. 5b, 7b, 8b, and 10b. When the contribution ($\Delta N_i$) of a specific ice source is negative, it indicates that incorporating this ice source leads to a reduction in total $N_i$, therefore suggesting a competitive interaction with other ice sources. In our simulations, the $\Delta N_i$ values for ice sources in the LP05 scheme are generally more negative than those in the K22 scheme. Thus, the assessment that competition among ice sources is stronger in the LP05 scheme is reasonable. To enhance clarity, we have provided further explanations to the related paragraphs to assist readers in understanding this point (line 513-518).

"Fig. 5b shows the differences in simulated $N_i$ between the reference experiments (OGW) and sensitivity experiments (no_OGW, no_DET and no_TKE). Larger differences in simulated $N_i$ between sensitivity experiments and the reference experiments indicate a more significant contribution from a respective ice crystal source (OGW-induced, detrained, or turbulence-induced). Specifically, increase or decrease of microphysical properties in the sensitivity experiments compared to the reference experiments reveals how each source contributes to enhancing or inhibiting the overall ice number concentrations."

In addition, we thank the reviewer for highlighting the need for more descriptions of the two nucleation schemes and the associated competition mechanisms. In response, we have expanded Section 2.2 by adding Section 2.2.2 to introduce the LP05 scheme and Section 2.2.3 to compare the two schemes for treating the competition between different ice sources.

We also appreciate the reviewer's suggestion to clarify how pre-existing ice crystals are treated in the two schemes, to ensure a fair comparison. The model simulates interactively the competition/interaction between new ice formation and pre-existing ice

(which are the ice crystals that are formed in previous time steps in the model grid or that transport/settle from other model grids). The fact that the LP05 scheme simulates a lower ice number concentration suggests that it contains a lower (not higher) concentration of pre-existing ice crystals than K22. Thus, the reduction in new ice formation is truly due to the strong competition treated in LP05.

Lastly, we are grateful for the reviewer's recommendation to elaborate on how changes in circulation influence subgrid-scale turbulence. In response, we have shown in Figure S3 subgrid-scale vertical velocity changes between the two schemes and refined the related explanations in the relevant sections of the manuscript.

The corresponding sentences have been revised as follows (in lines 411-420 and 817-820):

"Similarly, subgrid-scale vertical velocity increases in the K22_OGW-Climo experiment compared to the LP05_OGW-Climo experiment, particularly in the upper troposphere at mid- and high latitudes (Fig. S3). While these changes may enhance ice nucleation locally, their overall impact on $N_i$ remains limited, as vertical velocity changes are generally small (less than $\pm 0.002$ m s$^{-1}$) in most regions. Therefore, they are unlikely to explain the globally significant increase in $N_i$ simulated in the K22 scheme (Fig. 2)."

"An increase in cloud frequency may induce changes in global temperature, potentially affecting subgrid-scale vertical velocity, thereby impacting ice nucleation. However, these factors are not the key factors that cause the significant increase in $N_i$."

2. Since one purpose of this paper is to evaluate the K22 scheme, incorporating climatological (6 year) observational data is important for assessing the performance of both schemes. If obtaining global vertical profiles is challenging, bulk or regional observational data would still be valuable in determining whether K22 improves ice cloud simulations compared to LP05 from a climatological perspective.

Thank you very much for your comments. We agree that it is highly valuable to compare simulation results with observed cirrus clouds from a climatological perspective. However, it is challenging to get suitable observational data. We see this as an important direction for future work and may pursue it in a separate study. We will conduct further evaluations of the K22 scheme based on model climatology in our future studies by comparing modeled cirrus clouds with regional observational datasets (e.g., Krämer et al., 2016; 2020) and global satellite data as in Lyu et al. (2023).

The corresponding sentences have been added as follows (in lines 884-886):

"Further evaluations of the K22 scheme based on model climatology will be conducted in the future by comparing modelled cirrus with regional observational datasets (Krämer et al., 2016; Krämer et al., 2020) and global satellite data (Lyu et al., 2023)."

3. To deepen the insights of this study, the authors could discuss the potential impact of incorporating K22 into CAM6 on high cloud feedback. For example, if the proposed indirect mechanism for the higher ice number concentrations in K22 is true, large-scale circulation changes induced by global warming could modify sub-grid turbulence, subsequently affecting ice nucleation, cloud frequency, and longwave radiative effects.

Thank you very much for your comments. Investigating the potential impact of different nucleation schemes on high cloud feedback is indeed an important direction. In the revised manuscript, we have included discussions comparing cloud properties between the K22 and LP05 schemes, as shown in Fig. R2-1. Note that Fig. R2-1 has been included in the Supplementary Material. We have also added some discussions on the impacts of different nucleation schemes on cloud properties, which can modify global temperature, large-scale circulation, and sub-grid turbulence, subsequently affect ice nucleation, cloud frequency, and cloud radiative effects, and have important implications for high cloud feedbacks.

[Figure]

Figure R2-1. Annual mean differences between the K22_OGW-Climo and LP05_OGW-Climo experiments from 6-year climatological simulations (K22-LP05). Shown are grid-averaged ice number concentration ($N_i$) at 250 hPa, grid-averaged IWC at 250 hPa, grid-averaged ice effective radius (AREI) at 250 hPa, longwave cloud forcing (LWCF), shortwave cloud forcing (SWCF), and net cloud forcing (net CF). Meshed grid areas indicate values that are statistically significant at the 5% level.

When the nucleation scheme is switched from LP05 to K22, an increase in grid-averaged ice number concentration ($N_i$) is observed in the mid- and high- latitudes (Fig. R2-1a). Ice water content (IWC) also increases, particularly over high mountains (Fig. R2-1b), which may be attributed to enhanced depositional growth resulting from a greater number of smaller ice crystals in the K22 scheme.

The simulated ice effective radius (AREI) exhibits contrasting behavior over land and ocean in mid- and high latitudes (Fig. R2-1c). Over land, particularly mountainous areas, AREI tends to decrease, while over the ocean it increases. This suggests that compared to the LP05 scheme, the K22 scheme produces cirrus clouds with smaller ice crystals over elevated terrain and simulates larger crystals over oceanic regions.

In terms of longwave cloud forcing (LWCF), an increase is noted over high mountain areas in mid- and high latitudes (Fig. R2-1d). This is because the increase of $N_i$ over mountains in the K22 scheme. Interestingly, negative LWCF can be found over oceans at mid- and high latitudes. This phenomenon is primarily associated with the dominance of optically thin cirrus clouds formed via in-situ nucleation in these regions, as previously reported (Sassen & Cho, 1992; Sassen et al., 2008; Wang et al., 1996; Winker & Wielicki, 2010). The K22 scheme tends to enhance the spatial extent and occurrence frequency of such clouds. Over oceans, where vertical velocities are weaker than over land, these optically thin clouds become even thinner. This allows more longwave radiation to space, resulting in negative LWCF over oceans, consistent with the previous findings (Muri et al., 2014; Spang et al., 2024).

Shortwave cloud forcing (SWCF) increases in mid- and high latitudes (Fig. R2-1e), which can be attributed to increases in $N_i$, total cloud fraction and the contrast in shortwave albedo between the surface and clouds. The shortwave surface albedo of the ocean can reach 50-80 % under low solar angles (when the sun is near the horizon), while snow-covered land can exhibit albedos of 80-90 %. In contrast, cirrus clouds typically have a shortwave albedo of about 10-40 %. In addition, due to the low solar angles in high latitudes, cirrus clouds with a higher concentration of small ice crystals in K22 may enhance forward scattering rather than reflection, allowing more shortwave radiation to reach the surface. As a result, in high-latitude regions, the K22 scheme simulates less shortwave radiation reflected, leading to an increase in SWCF.

These paragraphs have been included in the revised manuscript (line 488-505).

    "When the ice nucleation scheme is switched from LP05 to K22, grid-averaged $N_i$ increases in the mid- and high latitudes (Fig. S13a). Ice water content (IWC) also increases (Fig. S13b) especially over high mountains. Ice effective radius (AREI) over land tends to be smaller and that over ocean tends to be larger, compared to the LP05 scheme (Fig. S13c). In mid- and high latitudes, longwave cloud forcing (LWCF) is increased over high mountains, as can be seen in Fig. S13d.  These changes can be explained by changes in the $N_i$ (Fig. S13h), as the K22 scheme generally simulates more ice crystals over high mountains. Interestingly, negative LWCF over oceans can be found in mid- and high latitudes. This phenomenon is primarily associated with the dominance of optically thin cirrus clouds formed via in-situ nucleation in these regions, as previously reported (Sassen & Cho, 1992; Sassen et al., 2008; Wang et al., 1996; Winker & Wielicki, 2010). The K22 scheme tends to enhance the spatial extent and occurrence frequency of

such clouds. Over oceans, where vertical velocities are weaker than over land, these optically thin clouds become even thinner. This allows more longwave radiation to space, resulting in negative LWCF over oceans, consistent with the previous findings (Muri et al., 2014; Spang et al., 2024). Shortwave cloud forcing (SWCF) increases in mid- and high latitudes (Fig. S13e), as the shortwave albedo of extensive cirrus clouds (10-40%) is lower than that of the underlying surface (ranging from 50-80% for oceans at low solar angles and 80-90% for snow-covered land). Changes in SWCF, LWCF and net cloud forcing (net CF) caused by the switch of ice nucleation scheme from LP05 to K22 are -0.51 W m$^{-2}$, 2.95 W m$^{-2}$, and 2.44 W m$^{-2}$, respectively. The change in the cloud radiative forcing may influence global temperature, which can modify large-scale circulation and sub-grid turbulence, subsequently affect ice nucleation, cloud frequency, and cloud radiative forcing, and have important implications for high cloud feedbacks (Murray & Liu, 2022)."

4. One structural issue in the manuscript is the overuse of short paragraphs, which disrupts the flow of the text. I recommend revisiting the paragraph structure and merging shorter paragraphs with logically related content to enhance readability and coherence. I will provide some specific suggestions in the minor comments, though they are not exhaustive.

Thank you very much for your thoughtful comments. In response, we have reorganized the paragraph structure and combined short paragraphs to improve the clarity and flow of the text.

Minor comments:

L7: I'd suggest reorganizing the abstract into two paragraphs or three at most.

Thank you very much for your valuable comments. In response, we have reorganized the abstract into two paragraphs for improved clarity and readability.

L88: Please give a brief reason why both field campaigns are used for validation. Any differences between these two or just for increasing the sample size?

Thank you very much for your comment. Observational data from field campaigns on cirrus clouds are generally considered to be more reliable compared to remote sensing methods such as satellite observations. We focus on the SPARTICUS and ORCAS campaigns in this study because they provide critical data on OGW-induced ice crystals. The SPARTICUS campaign involves flights over the mountainous regions from winter to summer, while the ORCAS campaign focuses on both ocean and continental regions during the summer. We added in the revised manuscript (line 243-246):

"We focus on the SPARTICUS and ORCAS campaigns in this study because they provide critical data on OGW-induced ice crystals. The SPARTICUS campaign involves

flights over the mountainous regions from winter to summer, while the ORCAS campaign focuses on both ocean and continental regions during the summer."

L98: No definition for "DET" upon its first appearance.

Thank you very much for your comment. We have added necessary definitions as follows (line 249-250):

"To isolate the effects of each source, we designed three sensitivity tests: no_DET (no detrainment), no_TKE (no CLUBB-TKE) and no_OGW (no OGWs),…"

L127: "thermaldynamic" to "thermodynamic"

Done. Thank you very much for your comment.

L150: Suggest moving this paragraph up.

Thank you very much for your comments. We have merged the paragraphs as suggested.

(line 208-214):

"Overall, the K22 scheme provides a more continuous and interactive treatment of multiple ice nucleation pathways, with a stronger emphasis on the dynamic interplay between supersaturation, aerosol concentrations, and pre-existing ice crystals. On the other hand, the LP05 scheme employs a stepwise approach that directly compares the potential for nucleation with the concentration of pre-existing ice crystals, imposing a threshold when nucleation occurs. Uncertainties exist regarding the relationship between the reduction of supersaturation and the suppression of nucleation caused by pre-existing ice crystals. This relationship and its impact on the number of nucleated ice crystals requires further investigations."

L152: "compared" to "compared to"

Corrected. Thank you.

L211: Since OGW-induced cloud nucleation is a very important source for ice cloud formation, I'd suggest comparing the climatology simulation results between land and oceans. The results over the land might be more contrasting between the two schemes.

Thank you very much for your insightful comments. The comparison of climatology simulation results between land and oceans is very important and is shown in Fig. 4e and

4f. We have included relevant discussions in the revised manuscript as suggested (line 372-383):

"Fig. 4 shows the global longitude-latitude distribution of annual mean $N_i$ at 250 hPa... Consistent with the results shown in Fig. 2, the K22_OGW-Climo experiment tends to produce higher ice number concentrations in all three types of simulated cirrus compared to the LP05_OGW-Climo experiment (Fig. 4a and 4b)…OGW-induced ice crystals in the K22 scheme are more abundant and broadly distributed over mountainous regions compared to the LP05 scheme (Fig. 4e and 4f). "

L215: Why more cirrus due to OGWs in high latitudes, particularly near the Poles (Figures 2e and 2f)?

Thank you very much for your comment. There are two reasons for this. First, there are many mountains and high plateaus in high latitudes, especially in Northern hemisphere and South Poles. These high lands are significant sources of orographic cirrus clouds. Second, other ice sources, such as convective detrainment and turbulence-induced ice crystals, are generally much weaker in high latitudes compared to low latitudes.

The relevant paragraph has been modified as follows (line 321-324):

"In both schemes, ice crystals detrained from convection are primarily concentrated in the tropical regions and mid-latitudes, and in situ nucleated ice crystals induced by turbulence are prevalent near the tropical tropopause layers (TTL) and in mid-latitudes. In contrast, due to the presence of mountains and high plateaus, orographic cirrus due to OGWs are concentrated over mid- and high latitudes."

L216: Please clarify the physical mechanisms for turbulence-induced ice nucleation.

Thank you very much for your comment. Turbulence-induced ice crystals are those ice crystals generated by vertical velocities from CLUBB-TKE. These vertical velocities result from small-scale turbulences with small Richarson numbers, which CLUBB is capable of capturing.

L232: "results" to "resulting", "resemble" to "resembles"

Done. Thank you.

L279: Any physical explanations for changes in sub-grid turbulence?

Thank you very much for your comment. The application of different ice nucleation schemes affects the formation of cirrus clouds in the simulation, which in turn influences temperature, leading to alternations in sub-scale turbulence.

L303: "Together with … (Fig. S7)" to "Together with simulated IWC and Dnum (Fig. S7)"

Thank you very much for your comments. We have modified the relevant sentence as suggested (line 509-510):

"Together with simulated IWC and $D_{num}$ (Fig. S14), both schemes produce results that generally agree with observational data. "

L316: What does "Temperature in X-axis" represent? Pressure-level mean temperature?

Thank you very much for your comment. We use temperature as the x-axis because ice nucleation is dependent on temperature. Also, temperatures correspond to specific pressure levels in the troposphere.

Thank you very much for your comments. We have modified the relevant sentence as suggested (line 540-541):

"Particularly in regions with temperatures below 215 K, where both schemes simulate their highest $N_i$ peaks, $\Delta N_i$ due to OGWs peaks positively at the corresponding temperatures."

L336-337: Be specific. It looks dependent on the source types.

Thank you very much for your comment. We have modified the relevant sentences as follows (line 543-551):

"Detrained and turbulence-induced $\Delta N_i$ values show different signs, fluctuating between positive and negative at different temperatures, indicating that the effects of the other two sources are uncertain and vary between the two schemes. In the LP05 scheme, detrained and turbulence-induced $\Delta N_i$ values are generally negative, suggesting that ice crystals from both detrainment and turbulence tend to inhibit $N_i$. In contrast, the K22 scheme exhibits varied signs of detrained and turbulence-induced $\Delta N_i$ values, with stronger fluctuations between positive and negative, indicating that these sources can either enhance or inhibit $N_i$. Notably, the positive $\Delta N_i$ values in detrained and turbulence-induced ice crystals are smaller in the LP05 scheme, suggesting stronger competition (inhibition effects) between ice sources in the LP05 scheme."

L351: As in K22 the detrained ice crystals do not have a significant competition, I'd expect that Dnum is slightly lower in K22_no_DET-SP (red lines in Fig. S9) than in

Thank you very much for your comment. As shown in Figure 5, the detrained ice in the K22 scheme has a very small effect (competition) on the overall $N_i$, thus the difference of Dnum between K22_no_DET-SP and K22_OGW-SP is small, and within the uncertain range (25th percentile to the 75th percentile) of the model simulations.

We have removed the related sentences to avoid confusion.

L388: Suggest moving this paragraph up

Thank you very much for your comment. We have merged the relevant paragraphs and re-written as follows (line 619-634):

"As shown in Fig. 7, multiple observed $N_i$ peaks correspond to different contributors to $\Delta N_i$, revealing that cirrus clouds exhibit multilayer structures with distinct ice sources. Simulated $N_i$ displays pronounced peaks above 225 K and near 210 K. At lower altitudes, where high $N_i$ values are observed at temperatures above 225 K, both schemes simulate positive $\Delta N_i$ values, indicating that ice crystals due to OGWs and detrainment are the dominant contributors to simulated $N_i$ in both schemes. In the LP05 scheme, turbulence-induced $\Delta N_i$ values are generally negative, implying that ice crystals from turbulence tend to suppress the overall $N_i$. In contrast, in the K22 scheme, turbulence-induced $\Delta N_i$ values fluctuate from negative to positive, suggesting inhibition between 215-230 K and enhancement of $N_i$ at temperatures $\geqslant$ 235 K. At the 210 K level, the overwhelmingly positive $\Delta N_i$ values due to turbulence in both schemes suggest that turbulence-induced ice crystals are the primary contributors to $N_i$ (Fig. 7b). However, in the LP05 scheme, $\Delta N_i$ values due to OGWs are negative, suggesting that OGW-induced ice crystals tend to inhibit ice crystal formation. In contrast, their impacts are minimal (~0) in the K22 scheme. In addition, both schemes simulate generally negative $\Delta N_i$ values due to detrainment, implying that detrained ice crystals tend to suppress the following ice formation."

L391: Why is its magnitude so large between 220 and 230 K?

Thank you very much for your comment. The contribution ($\Delta N_i$) of ice crystals generated by turbulence can reach up to -10 L$^{-1}$. While this may appear relatively large in the plot, it is not considered a large value.

L402: It seems like no_TKE experiment shows the highest peak. Please double check.

Thank you very much for your comment. We focused on the peaks in the OGW experiments (which include no-TKE experiments). We have modified the sentence to avoid any confusion as follows (line 654-656):

"Near 215 K, the OGW experiments in both schemes simulate high $N_i$ peaks that closely match the observed high peak near 218 K."

L413: Please rephrase this sentence.

Thank you very much for your comments. We have rephrased the sentence to address the issue more clearly as follows (line 674-675):

"In the lower part of cirrus clouds (T > 225K), negative $\Delta N_i$ values of all three ice crystal sources in the LP05 scheme suggest universal competition among these sources."

L456: "OGW-induce" to "OGW-induced"

Done. Thank you.

L456: Are these dates selected when the simulations of both schemes align with the observations?

Thank you very much for your comment. The median $N_i$ values from OGW experiments in both schemes fall within the observed $N_i$ range. In addition, the primary contributor of ice crystals on these selected dates is the OGWs.

L481: "considering" to "with a focus on"

Done. Thank you.

L490-495: Are there any physical reasons, or formula-related proofs? If not, the first reason is more like an assumption.

Thank you very much for your comments. The competition between ice sources in both the LP05 and K22 schemes is based on certain assumptions in the schemes. We modified the sentences as follows (line 806-813):

"The direct reason lies in their different assumptions of treating the competition between pre-existing ice and nucleated ice crystals. The K22 scheme emphasizes the dynamic interplay between supersaturation, aerosol concentrations and pre-existing ice, allowing homogeneous nucleation, heterogeneous nucleation and the growth of pre-existing ice crystals to occur simultaneously. In contrast, the LP05 scheme is based on an empirical framework that favors a specific nucleation pathway. In the LP05 scheme, heterogeneous nucleation is favored at low supersaturation and high INP concentrations, while homogeneous nucleation dominates at high supersaturations. Pre-existing ice crystals consume supersaturation before new ice nucleation can occur. This may result in a stronger competition in the LP05 scheme, suppressing homogeneous nucleation. "

L496-500: Have you examined the changes in large-scale circulations and their association with sub-grid turbulence variations? If not, you would have to soften your tone when proposing the indirect reason.

Thank you very much for your valuable comments. We have softened our tone when we talk about the indirect reason. The relevant paragraph has been modified as follows (line 814-820):

"The indirect reason is related to the increase in ice number concentrations within the K22 scheme, which appears to lead to higher cloud frequency. This can be due to the presence of smaller ice crystals in the K22 scheme, which have lower fall speeds, allowing them to travel over broader regions before completely sublimated. An increase in cloud frequency may subtly induce changes in global temperature, potentially affecting turbulence and subgrid-scale vertical velocity, thereby impacting ice nucleation. However, these factors are not the key factors that cause the significant increase in $N_i$."

L524: Please move it to the last paragraph.

Thank you very much. We have merged the relevant paragraphs.

L532: Be specific for these critical INPs.

Thank you very much for your comments. We have modified the relevant sentence as follows (line 879-880):

"We note that other critical INPs (such as black carbon, metallic particles, biological materials) besides mineral dust are not currently represented in current ice nucleation schemes (Lin et al., 2025)."

**Reference**

Krämer, M., Rolf, C., Luebke, A., Afchine, A., Spelten, N., Costa, A., Meyer, J., Zöger, M., Smith, J., Herman, R. L., Buchholz, B., Ebert, V., Baumgardner, D., Borrmann, S., Klingebiel, M., & Avallone, L. (2016). A microphysics guide to cirrus clouds – Part 1: Cirrus types. *Atmos. Chem. Phys.*, *16*(5), 3463-3483. https://acp.copernicus.org/articles/16/3463/2016/

Krämer, M., Rolf, C., Spelten, N., Afchine, A., Fahey, D., Jensen, E., Khaykin, S., Kuhn, T., Lawson, P., Lykov, A., Pan, L. L., Riese, M., Rollins, A., Stroh, F., Thornberry, T., Wolf, V., Woods, S., Spichtinger, P., Quaas, J., & Sourdeval, O. (2020). A microphysics guide to cirrus – Part 2: Climatologies of clouds and humidity from observations. *Atmos. Chem. Phys.*, *20*(21), 12569-12608. https://acp.copernicus.org/articles/20/12569/2020/

Lin, L., Liu, X., Zhao, X., Shan, Y., Ke, Z., Lyu, K., & Bowman, K. P. (2025). Ice nucleation by volcanic ash greatly alters cirrus cloud properties. *Science Advances*, *11*(19), eads0572. https://doi.org/doi:10.1126/sciadv.ads0572

Lyu, K., Liu, X., Bacmeister, J., Zhao, X., Lin, L., Shi, Y., & Sourdeval, O. (2023). Orographic Cirrus and Its Radiative Forcing in NCAR CAM6. *Journal of Geophysical Research: Atmospheres*, *128*(10), e2022JD038164. https://agupubs.onlinelibrary.wiley.com/doi/abs/10.1029/2022JD038164

Muri, H., Kristjánsson, J. E., Storelvmo, T., & Pfeffer, M. A. (2014). The climatic effects of modifying cirrus clouds in a climate engineering framework. *Journal of Geophysical Research: Atmospheres*, *119*(7), 4174-4191. https://doi.org/https://doi.org/10.1002/2013JD021063

Murray, B. J., & Liu, X. (2022). Chapter 15 - Ice-nucleating particles and their effects on clouds and radiation. In K. S. Carslaw (Ed.), *Aerosols and Climate* (pp. 619-649). Elsevier. https://doi.org/https://doi.org/10.1016/B978-0-12-819766-0.00014-6

Sassen, K., & Cho, B. S. (1992). Subvisual-Thin Cirrus Lidar Dataset for Satellite Verification and Climatological Research. *Journal of Applied Meteorology and Climatology*, *31*(11), 1275-1285. https://doi.org/https://doi.org/10.1175/1520-0450(1992)031<1275:STCLDF>2.0.CO;2

Sassen, K., Wang, Z., & Liu, D. (2008). Global distribution of cirrus clouds from CloudSat/Cloud-Aerosol Lidar and Infrared Pathfinder Satellite Observations (CALIPSO) measurements. *Journal of Geophysical Research: Atmospheres*, *113*(D8). https://doi.org/https://doi.org/10.1029/2008JD009972

Spang, R., Müller, R., & Rap, A. (2024). Radiative effect of thin cirrus clouds in the extratropical lowermost stratosphere and tropopause region. *Atmos. Chem. Phys.*, *24*(2), 1213-1230. https://doi.org/10.5194/acp-24-1213-2024

Wang, P.-H., Minnis, P., McCormick, M. P., Kent, G. S., & Skeens, K. M. (1996). A 6-year climatology of cloud occurrence frequency from Stratospheric Aerosol and Gas Experiment II observations (1985–1990). *Journal of Geophysical Research: Atmospheres*, *101*(D23), 29407-29429. https://doi.org/https://doi.org/10.1029/96JD01780

Winker, D. M. P. J. C. J. A. A. S. A. C. R. J. C. P. R. F. P. F. Q. H. R. M., & Wielicki, B. A. (2010). The CALIPSO mission. *Bulletin of the American Meteorological Society*, *91(9), 1211–1230. , DOI =* https://doi.org/10.1175/2010BAMS3009.1.

---

## Author Response (AR2)

We sincerely thank the two anonymous reviewers for their thoughtful and constructive comments. In the following, we provide detailed responses to each comment and indicate how they have been addressed in the revised manuscript. The reviewers' comments are shown in blue, our replies are in black, and the corresponding revisions in the manuscript are highlighted in red (line numbers refer to the tracked version of the revised manuscript).

Recommendation: Return to Authors for Major Revisions

**Overview**

This manuscript has improved compared to previous versions—thank you for your continued efforts. However, the presentation still lacks clarity and logical flow, particularly in key technical sections. Below, I provide a set of major and minor comments that should be addressed to improve the overall quality and readability of the paper.

Major comments/questions:

1. Clarification and Completeness of the K22 Scheme Description

The presentation of the K22 scheme still requires significant clarification. Several key conceptual and technical aspects remain unclear:

• Scope of K22: The K22 scheme is designed for cirrus clouds. To my understanding, it does not apply to freezing processes in mixed-phase clouds. Please clarify this explicitly.

Reply: Thank you very much for your helpful and constructive comments. Yes, the K22 scheme is designed only for cirrus clouds and does not apply to freezing processes in mixed-phase clouds.

In CAM6, cirrus clouds are defined as the clouds with temperatures below -37 °C and mixed-phase clouds are defined as the clouds with temperatures between 0 and -37 °C. The freezing processes in cirrus clouds and mixed-phase clouds are treated differently. Heterogeneous ice nucleation in mixed-phase clouds is based on the classical nucleation theory including immersion, deposition and contact freezing with rates depending on the properties of mineral dust and black carbon aerosols (Hoose et al., 2010; Wang et al., 2014).

The relevant paragraph has been modified as follows (line 92-96):

"In CAM6, cirrus clouds are defined as the clouds with temperatures below -37 °C and mixed-phase clouds are defined as the clouds with temperatures between 0 and -37 °C. Ice nucleation in cirrus clouds is treated differently (see section 2.2) from that in mixed-phase clouds. Ice nucleation in mixed-phase clouds is treated based on the classical nucleation theory including immersion, deposition and contact freezing with rates

depending on the properties of mineral dust and black carbon aerosols (Hoose et al., 2010; Wang et al., 2014). "

• Equations (1) – (5): These equations appear to describe heterogeneous ice nucleation only. If so, why is homogeneous freezing not included in this section? Please incorporate the corresponding parameterization for homogeneous nucleation. Furthermore, how is INP activation efficiency for homogeneous freezing quantified?

Reply: Following the reviewer's comment, we have provided an explanation of homogeneous nucleation in the revised manuscript. Please note that homogeneous nucleation refers to the spontaneous freezing of aerosol (sulfate) solution droplets, and does not involve INPs (e.g., dust, black carbon). The relevant paragraph has been modified as follows (line 100-115):

"In the K22 parameterization, the number of activated solution droplets ( $n_{\text{homo}}$ ) over time is calculated based on the freezing rate (j), following the expression:

$$n_{homo} = n_{sulfate}[1 - exp(\int -jdt)] \tag{1}$$

 $n_{sulfate}$  is the initial number concentration of sulfate solution droplets, the freezing rate j is determined using the liquid water volume (V) of the solution droplet population and a rate coefficient (J) derived from a water activity-based formula (Koop et al., 2000) (j=VJ). The parameterization assumes a monodisperse size distribution of solution droplets with radius of 0.25 µm, neglecting the presence of a small amount of soluble material in the droplets. Vertical velocity (w), supersaturation with respect to ice ( $S_i$ ), and temperature (T) significantly influence the water activity so that  $J=J(w,S_i,T)$  (Baumgartner et al., 2022; Kärcher et al., 2022; Liu & Penner, 2005). The thermodynamic threshold  $S_{hom}$  for homogeneous freezing to take place is estimated through an iterative process in which the deposition growth of ice crystals from previously frozen solution droplets reduces the supersaturation. This quenching process is a function of T, w, and the mean droplet size (Kärcher et al., 2022). Once  $S_{hom}$  is determined, the number concentration of newly nucleated ice crystals is computed using  $S_{hom}$ ,  $S_i$  and effective updraft speed (see Equation (6) below). More detailed information can be found in Kärcher et al. (2022)."

 Supersaturation and Equation (2): Equation (2) relies on supersaturation (S) to compute Φ. However, S is determined from Equation (3), which seems to be highly nonlinear in S. What numerical method or approximation is used to solve Equation (3)?

Reply: The ice supersaturation threshold at heterogeneous activation-relaxation is determined by numerical iteration when the dS/dt=0 (i.e., the production and loss of supersaturation in equation (3) (now equation (4)) are equal) and used to compute the  $\Phi$  from INPs. If homogeneous nucleation also occurs, the ice supersaturation threshold at

homogeneous activation-relaxation determined similarly is used to compute the  $\Phi$  from INPs.

We added a sentence (line 155-158):

"The ice supersaturation threshold at heterogeneous activation-relaxation is determined by numerical iteration when the dS/dt=0 (i.e., the production and loss of supersaturation in equation (4) are equal) and used to compute the  $\Phi$  from INPs in equation (3). If homogeneous nucleation also occurs, the ice supersaturation threshold at homogeneous activation-relaxation determined similarly is used to compute the  $\Phi$  from INPs."

• Role of Vertical Velocity: The treatment of vertical velocity is unclear. My understanding is that an effective vertical velocity weff is derived from Equation (5) and passed into Equation (3), with a steady-state assumption dSdt=0. If so, how are w(q,het) and w(q,pre) derived? Is Equation (4) used in this context? Please specify how the the right-hand side of Equation (4) are quantified.

Reply: Yes, weff derived from equation (5) (now equation (6)) is used to calculate ice number from homogeneous nucleation. To calculate w(q,het), the loss term due to the deposition of water vapor onto ice crystals formed from heterogeneous nucleation, denoted as  $L_{q,het}$ , must first to be determined:

$$L_{q,het} = \sum_{k=1}^{K} \frac{n_k}{n_{sat}} \frac{dN_k}{dt}$$

where the index k denotes an INP class, with associated ice number concentrations  $n_k$  resulting from nucleation of the fraction of INPs that become ice-active within a supersaturation interval  $\Delta S_k$ .  $N_k$  is the number concentration of water molecules per ice crystal formed from INPs in each supersaturation class. The water molecule number concentration at ice saturation  $n_{\text{sat}}$  is obtained from Murphy and Koop (2005). The rate of change in the number of water molecules per ice crystal is given by  $\frac{dN_k}{dt} = 4\pi r_k D_k n_{\text{sat}} S$ , where  $r_k$  is ice crystal radii, assuming a spherical volume centered on the INP core:  $r_k = (r_c^3 + \frac{vN_k}{4\pi/3})^{1/3}$ . Here, v is the volume of a single water molecule in ice and  $r_c$  is the radius of the dry aerosol particle core (assumed to be 0.2 µm). The effective diffusivity  $D_k$  is given by:  $D_k = D_v(\frac{r_k}{r_k+l} + \frac{d}{\alpha_k r_k})^{-1}$ , where  $D_v$  is the water diffusion coefficient in air, l is the jump distance for water molecules (approximately equal to the mean free path),  $d=4D_v/v$  is the diffusion length scale, with v being the mean thermal speed of water, and  $\alpha_k$  is the deposition coefficient specific to the ice crystals formed in the supersaturation interval  $\Delta S_k$ .

The quenching velocity due to heterogeneous nucleation w(q,het) is then calculated as:

$$w(q, het) = \frac{L_{q,het}}{a(S+1)}$$

To compute the quenching velocity due to pre-existing ice, w(q,pre), the loss term due to the removal of water vapor onto pre-existing ice crystals  $L_{q,pre}$  is calculate as:

$$L_{q,pre} = \int_0^s \frac{4\pi}{v n_{sat}} \frac{dn}{ds'} \left( \int_{\tau(s')}^{t(s)} r^2 \frac{dr}{dt} dt \right) ds'$$

Finally, w(q,pre) is:

$$w(q, pre) = \frac{L_{q,pre}}{a(S+1)}$$

The relevant paragraph has been modified as follows (line 137-158):

"When  $\frac{ds}{dt} = 0$  in Equation (4), we can define the quenching velocity  $w_{q,pre}$  due to pre-existing ice crystals as:

$$w_{q,pre} = \frac{\int_{0}^{s} \frac{4\pi}{v n_{sat} ds'} (\int_{\tau(s')}^{t(s)} r^{2} \frac{dr}{dt} dt) ds'}{a(s+1)},$$
 (5)

where the loss term of water vapor includes the contribution from pre-existing ice. The quenching velocity due to heterogeneous ice nucleation  $w_{q,het}$  can be calculated similarly based on Kärcher et al. (2022), using the equation:  $w_{q,het} = \frac{L_{q,het}}{q(s+1)}$ . Here,  $L_{q,het}$  is the loss term due to the deposition of water vapor onto ice crystals formed from heterogeneous nucleation:  $L_{q,het} = \sum_{k=1}^{K} \frac{n_k}{n_{sat}} \frac{dN_k}{dt}$ . The index k denotes an INP class, with corresponding ice number concentrations  $n_k$  that result from nucleation of the fraction of INPs that become ice-active within a supersaturation interval  $\Delta S_k$ .  $N_k$  represents the number concentration of water molecules per ice crystal formed from INPs in each supersaturation class. The water molecule number concentration at ice saturation  $n_{\text{sat}}$  is obtained from Murphy and Koop (2005). The rate of change in the number of water molecules per ice crystal is given by  $\frac{dN_k}{dt} = 4\pi r_k D_k n_{sat} S$ , where  $r_k$  is ice crystal radii, assuming a spherical volume centered on the INP core:  $r_k = (r_c^3 + \frac{vN_k}{4\pi/3})^{1/3}$ . In this expression, v is the volume of a single water molecule in ice, and  $r_c$  is the radius of the dry aerosol core (assumed to be 0.2  $\mu$ m). The effective diffusivity  $D_k$  is given by:  $D_k =$  $D_v(\frac{r_k}{r_k+l}+\frac{d}{\alpha_k r_k})^{-1}$ , where  $D_v$  is the water diffusion coefficient in air, l is the jump distance for water molecules (approximately equal to the mean free path),  $d = 4D_v/v$  is the diffusion length scale, v is the mean thermal speed of water molecules, and  $\alpha_k$  is the deposition coefficient specific to ice crystals formed within the supersaturation interval  $\Delta S_{\rm k}$ ."

• Homogeneous Freezing – Liquid Water Volume (V): Line 94 refers to a required liquid water volume V. How is this calculated? Is it based on an assumed droplet

size (e.g.,  $0.25~\mu m$  as mentioned in Line 97) and estimated droplet number concentration? Please describe the approach in detail.

Reply: The droplet population is assumed to be a lognormal droplet distribution with modal radius of 0.25  $\mu$ m, and geometric standard deviation of 1 (i.e., monodisperse). V is the volume of a solution droplet. The droplet number concentration is assumed to be 500 per cubic centimeter. The estimated activated droplet number concentration is calculated by  $n_{homo} = n_{sulfate}[1 - exp(\int -jdt)]$ .

The relevant paragraph has been modified as follows (line 105-107):

"The parameterization scheme assumes a monodisperse size distribution of solution droplets with radius of 0.25  $\mu$ m, neglecting the presence of a small amount of soluble material in the droplets."

Please address each of the above points clearly and systematically. These questions are fundamental for assessing whether the authors have a solid understanding of the K22 scheme and its implementation. Additional technical details can be placed in the Supporting Information if necessary.

**2. Analysis of Freezing Frequencies and Vertical Velocity Contributions**

It would significantly strengthen the manuscript if the authors analyzed the relative frequencies of homogeneous versus heterogeneous freezing events predicted by the K22 scheme. Specifically:

- Show the distribution of weff (from Eq. 5) as a function of latitude and altitude.
- Examine the role of orographic gravity waves (OGW) in producing positive weff values.
- Clarify whether w(q,het) is calculated as w(q,het)=w-w(q,pre). If so, provide the frequency and latitudinal/vertical distribution of w(q,het).

This analysis will offer deeper insight into the competition between homogeneous and heterogeneous freezing. Currently, the paper only discusses changes in ice crystal number concentration (Ni), but the vertical velocity distribution—being the root cause of these changes—deserves direct analysis.

Reply: Thank you very much for your helpful and constructive comments. It is a good idea to check homogeneous and heterogeneous freezing frequencies in the K22 scheme based on the effective vertical velocity  $w_{\text{eff}}$  and quenching speed due to heterogeneous nucleation.

• The distribution of  $w_{\text{eff}}$  (from Eq. 6, formerly Eq. 5) in the K22\_OGW-Climo experiment as a function of latitude and altitude is shown in Fig. R1-1. The figure is included in the supplemental figures. The relevant paragraph has been modified as follows (line 302-304):

"In the K22\_OGW-Climo experiment, strong  $w_{\text{eff}}$  is found over mid- and high latitudes (Fig. S1), with the large positive  $w_{\text{eff}}$  occurring primarily over the high mountain regions (Fig. S2). This pattern indicates the important contribution of OGWs in producing positive  $w_{\text{eff}}$  values."

Figure R1-1. Annual mean  $w_{\text{eff}}$  (unit: m s-1) as a function of latitude and altitude from the K22-OGW-Climo experiment.

• Figure R1-1 shows strong  $w_{\text{eff}}$  over mid- and high latitudes. In addition,  $w_{\text{eff}}$  at 250 hPa is presented in Fig. R1-2. The large positive  $w_{\text{eff}}$  over mid- and high latitudes is primarily located over the high mountains, indicating the contributing role of OGWs in producing positive  $w_{\text{eff}}$  values. The  $w_{\text{eff}}$  spatial distribution indicates the frequency of homogeneous nucleation occurrences.

Figure R1-2. Annual mean  $w_{\text{eff}}$  at 250 hPa from the K22-OGW-Climo experiment (unit: m s-1).

• The quenching speed w(q,het) is not equal to w - w(q,pre) in most cases because the effective updraft speed is defined as  $w_{eff}=w - w(q,het) - w(q,pre)$ . The zonal mean w(q,het) is shown in Fig. R1-3, which indicates that strong quenching effect due to heterogeneous ice nucleation primarily occurs in the lower troposphere, especially in the Northern Hemisphere and in the mid-latitudes of the Southern Hemisphere. High concentrations of coarse mode dust are found in the lower troposphere, especially in mid-latitudes (Fig. R1-4). Fig. R1-5 shows w(q,het) and coarse mode dust number concentrations at 350 hPa. A pronounced concentration coarse mode dust is found over Tibetan Plateau, corresponding one high value region of w(q,het). This suggests that elevated coarse mode dust number concentrations are necessary for the occurrence of strong w(q,het).

Figure R1-3. Zonal mean  $w(q_het)$  in the K22-OGW-Climo experiment (Unit: m s-1).

Figure R1-4. Zonal mean coarse mode dust number concentration in the K22-OGW-Climo experiment (Unit: L-1).

Figure R1-5.  $w(q_het)$  and coarse mode dust at 350 hPa in the K22-OGW-Climo experiment.

**Minor comments/questions:**

1. Lines 1–2: If the study explores all ice formation pathways, sources like detrainment should be mentioned. The title currently suggests the focus is limited to INP activation. Consider emphasizing "Ice Crystal Formation from INP Activation in Cirrus Clouds" for clarity.

Thank you very much for your constructive comment. Since this study focuses on ice nucleation in cirrus clouds by using two ice nucleation schemes, we would like to keep the current title. However, we modified the "ice nucleation" to "ice crystal formation" for the consideration of all ice formation pathways in the first sentence in the paragraph as follows (line 7-8):

"Ice crystal formation in cirrus clouds is poorly understood, and its representation remains a challenge in global climate models."

2. Lines 18–20: The sentence beginning with "However..." is unclear and should be rewritten for better readability.

Thank you very much for your comment. The relevant sentence has been modified as follows (line 19-21):

- "Due to its distinct competition parameterizations, the K22 scheme exhibits less contribution from minor ice sources (convection detrained and turbulence-induced)."
- 3. Lines 20–22: The impact summary is too weak. This study could help clarify regional and dynamical controls on INP activation. Please strengthen the impact statement.

Thank you very much for your constructive comment. The relevant paragraph has been modified as follows (line 21-24):

- "This underscores the significance of competition mechanisms within ice nucleation schemes and helps clarify regional and dynamical controls on ice sources in cirrus clouds."
- 4. Lines 25–26: The claim that cirrus clouds "warm the planet" needs more explanation. While cirrus do absorb longwave radiation, the extent to which they warm the surface depends on their net radiative forcing. Please clarify.

Thank you very much for your helpful and constructive comment. The relevant paragraph has been modified as follows (line 27-28):

- "They can also absorb terrestrial longwave radiation, thereby contributing to warming the atmosphere."
- 5. Lines 26–27: The phrase "cirrus clouds determine cloud radiative forcing" is too strong. Other cloud types (e.g., stratocumulus) also contribute significantly. Please moderate the statement.

Thank you very much for your helpful and constructive comment. The relevant paragraph has been modified as follows (line 28-30):

- "The balance between these two opposite processes is greatly influenced by the microphysical properties of ice crystals in cirrus clouds, which in turn affects the net cloud radiative forcing."
- 6. Lines 58–62: Please clarify the distinction between uncertainty in supersaturation and INP activation efficiency. In my view, supersaturation is an external factor affecting INP efficiency (which also depends on chemical composition). The aerosol number concentration determines the number of INPs. What exactly is meant by "activation efficiency" in this context as an independent factor for INP activation?

Thank you very much for your helpful and constructive comment. By the activation efficiency we meant activation fraction at a given condition, which depends on physical and chemical properties (e.g., morphology, chemical composition) of INPs. The relevant sentence has been modified as follows (line 60-63):

"Limited knowledge of the number concentration and properties (e.g., morphology, chemical composition) of INPs in the upper troposphere complicates the model prediction of cirrus clouds microphysical properties (Kärcher et al., 2022; Knopf & Alpert, 2023)."

7. Lines 68–70: Does the prior sentence imply that the K22 scheme is already implemented in CAM6? Please confirm.

Thank you very much for your helpful and constructive comment. Sorry that it was not clear. The K22 scheme is implemented in CAM6 for the first time in this study. The relevant sentence has been modified as follows (line 68-71):

"In this study, a new parameterization (Kärcher, 2022), referred to as K22, that encompasses homogeneous nucleation, heterogeneous nucleation, their interactions, and competition with preexisting cirrus ice, is integrated into CAM6. We further evaluate its effects on ..."

8. Lines 85–88: I use CESM as well but am not familiar with INP-MAM4 interactions as described. Could you specify the CESM code version and point to the relevant code section? Also, conceptually, converting INPs to cloud borne aerosols seems problematic, as cloud-borne aerosols typically refer to CCN, not IN. I may be mistaken, but showing the code would clarify this.

Thank you very much for your helpful and constructive comment. In CESM2, cloud borne aerosols have been extended from CCN in warm clouds to including INPs for ice nucleation in cold clouds. Upon ice nucleation, INPs will be converted to cloud-borne aerosols as well. Future model development could separate "cloud borne aerosols" into "droplet-borne aerosols" for warm clouds and "ice-borne aerosols" for cold clouds.

The version of CESM used in this study is CESM2.2.0. The relevant code section can be found in components/cam/src/physics/cam/nucleate\_ice\_cam.F90.The source code means that the INPs are converted to cloud borne once they are nucleated.

```
! Move aerosol used for nucleation from interstial to cloudborne,
! otherwise the same coarse mode aerosols will be available again
! in the next timestep and will supress homogeneous freezing.
if (prog_modal_aero .and. use_preexisting_ice) then
if (separate_dust) then
call endrun('nucleate_ice_cam: use_preexisting_ice is not supported in separate_dust mode (MAM7)')
endif
ptend%q(i,k,cnum_idx) = -(odst_num * icldm(i,k))/rho(i,k)/1e-6_r8/dtime
cld_num_coarse(i,k) = cld_num_coarse(i,k) + (odst_num * icldm(i,k))/rho(i,k)/1e-6_r8
```

```
ptend\%q(i,k,cdst\_idx) = - \ odst\_num \ / \ dst\_num \ * \ icldm(i,k) \ * \ coarse\_dust(i,k) \ / \ dtime cld\_coarse\_dust(i,k) = cld\_coarse\_dust(i,k) + odst\_num \ / \ dst\_num \ * icldm(i,k) \ * \ coarse\_dust(i,k) end \ if "
```

9. Lines 97–98: Please include the homogeneous freezing parameterization in the paper. Especially, how is the liquid water content (V) calculated, and what assumptions are made (e.g., droplet size =  $0.25 \mu m$ )? How is droplet number determined? This will help readers follow without referring back to Karcher et al. (2022).

Please see our response to your previous major comment above. V is the volume of a solution droplet. The droplet number concentration is assumed to be 500 per cubic centimeter. The estimated activated droplet number concentration is calculated by  $n_{homo} = n_{sulfate}[1 - exp(\int -jdt)]$ .

10. Lines 114–115: How exactly is water vapor removed? Is it due to deposition on preexisting ice, newly formed ice, or other processes like entrainment/detrainment? Please clarify and list the quantification of all the terms.

Thank you very much for your helpful and constructive comment. The water vapor is removed by deposition onto newly nucleated ice or onto pre-existing ice (i.e., ice formed from previous time steps). The relevant paragraph has been modified as follows (line 130-131):

"The removal of water vapor can be caused by the deposition onto newly nucleated ice crystals or onto pre-existing ice crystals."

11. Lines 150–152: I still find the competition between homogeneous, heterogeneous, and pre-existing ice formation difficult to follow. Is this primarily reflected in Equation (5)? If weff<0, does that imply no homogeneous freezing?

Thank you very much for your helpful and constructive comment. The K22 scheme represents the competition between homogeneous, heterogeneous and pre-existing ice by quenching velocities, reflected in Eq. (5) (now Eq. (6)). We have added the equations for calculating the quenching velocities from heterogeneous nucleation and from pre-existing ice in the revised manuscript. Please see our response to your major comment #1 above.

Yes. If  $w_{\text{eff}} < 0$ , this means that no homogeneous freezing happens.

12. Lines 179–181: Were these simulations free-running or nudged? If they were free-running, I'm concerned that large-scale meteorological differences may limit meaningful comparison with field campaign observations.

Thank you very much for your comment. The 6-year climatology simulations are freerunning, while the simulations related to two flight campaigns are wind (UV)-nudged towards the MERRA2 reanalysis. We added a sentence in the revised manuscript (line 221-222):

"For the nudged simulations for the two field campaigns (Table 2), the modelled horizontal winds are nudged towards the MERRA2 reanalysis data."

**Reference:**

- Baumgartner, M., Rolf, C., Grooß, J. U., Schneider, J., Schorr, T., Möhler, O., Spichtinger, P., & Krämer, M. (2022). New investigations on homogeneous ice nucleation: the effects of water activity and water saturation formulations. *Atmos. Chem. Phys.*, 22(1), 65-91. <a href="https://doi.org/10.5194/acp-22-65-2022">https://doi.org/10.5194/acp-22-65-2022</a>
- Kärcher, B. (2022). A Parameterization of Cirrus Cloud Formation: Revisiting Competing Ice Nucleation. *Journal of Geophysical Research: Atmospheres*, *127*(18), e2022JD036907. <a href="https://doi.org/10.1029/2022JD036907">https://doi.org/https://doi.org/10.1029/2022JD036907</a>
- Kärcher, B., DeMott, P. J., Jensen, E. J., & Harrington, J. Y. (2022). Studies on the Competition Between Homogeneous and Heterogeneous Ice Nucleation in Cirrus Formation. *Journal of Geophysical Research: Atmospheres*, *127*(3), e2021JD035805. https://doi.org/https://doi.org/10.1029/2021JD035805
- Knopf, D. A., & Alpert, P. A. (2023). Atmospheric ice nucleation. *Nature Reviews Physics*, 5(4), 203-217. <a href="https://doi.org/10.1038/s42254-023-00570-7">https://doi.org/10.1038/s42254-023-00570-7</a>
- Koop, T., Luo, B., Tsias, A., & Peter, T. (2000). Water activity as the determinant for homogeneous ice nucleation in aqueous solutions. *Nature*, 406(6796), 611-614. https://doi.org/10.1038/35020537
- Liu, X., & Penner, J. (2005). Ice nucleation parameterization for global models. *Meteorologische Zeitschrift*, 14, 499-514, DOI = https://doi.org/410.1127/0941-2948/2005/0059.
- Murphy, D. M., & Koop, T. (2005). Review of the vapour pressures of ice and supercooled water for atmospheric applications. *Quarterly Journal of the Royal Meteorological Society: A journal of the atmospheric sciences, applied meteorology and physical oceanography*, *131*(608), 1539-1565, ISSN = 0035-9009, DOI = https://doi.org/1510.1256/qj.1504.1594.

We sincerely thank the two anonymous reviewers for their thoughtful and constructive comments. In the following, we provide detailed responses to each comment and indicate how they have been addressed in the revised manuscript. The reviewers' comments are shown in blue, our replies are in black, and the corresponding revisions in the manuscript are highlighted in red (line numbers refer to the tracked version of the revised manuscript).

Thanks to the authors for addressing the reviewer comments. I recommend accepting the manuscript after the following minor issues are resolved.

1. L19: The phrase "less contribution" seems to better fit the context than "less competition". The authors may consider rewording it.

Thank you very much for your suggestion. We reworded "less competition" to "less contribution". The relevant paragraph has been modified as follows (line 19-21):

"Due to its distinct competition parameterizations, the K22 scheme exhibits less contribution from minor ice sources (convection detrained and turbulence-induced)."

2. L391-392: Should the positions of "SWCF" and "LWCF" be swapped? Figures R2-1 (d) and (e) show that the average LWCF and SWCF are -0.506 and 2.950 W/m2, respectively. Please check.

Thank you very much for your comment. We have carefully re-checked SWCF and LWCF. The SWCF and LWCF values from both the K22 and LP05 schemes are presented in Figure R2-1. The distributions of SWCF and LWCF appear similar between both schemes. Interestingly, the differences in SWCF and LWCF changes between the two schemes contradict the conventional expectation that more frequent cirrus clouds in K22 compared to LP05 should result in a stronger positive LWCF and a more negative SWCF. These unexpected results, derived from the RRTMG radiation scheme, are likely influenced by the presence of thin cirrus clouds and the high solar zenith angles typical of mid- and high-latitude regions. The cirrus clouds in question are sufficiently thin to allow longwave radiation to pass through. Additionally, the high solar zenith angles may enhance the scattering of shortwave radiation within ice crystals, allowing more shortwave radiation to reach the surface instead of being reflected back to space. Moreover, the surface reflectivity may also play an important role. At high solar zenith angles, ocean surfaces can reflect 50-80% of incoming solar radiation, and snow-covered land can reflect 80-90% sunlight, while cirrus clouds can only reflect 10-40%. These combined factors may explain the unexpected results of positive change of SWCF and negative change in LWCF from LP05 to K22 schemes.

Figure R2-1. The SWCF and LWCF in both the K22 (upper panel) and LP05 schemes (lower panel).